

# On weak ergodicity breaking in mean-field spin glasses

**Giampaolo Folena[1] and Francesco Zamponi[2]**

**1** Department of Chemistry, Duke University, Durham, North Carolina 27708, USA
**2** Laboratoire de Physique de l'Ecole Normale Supérieure, ENS, Université PSL, CNRS,
Sorbonne Université, Université de Paris, F-75005 Paris, France

## Abstract

The weak ergodicity breaking hypothesis postulates that out-of-equilibrium glassy systems lose memory of their initial state despite being unable to reach an equilibrium stationary state. It is a milestone of glass physics, and has provided a lot of insight on the physical properties of glass aging. Despite its undoubted usefulness as a guiding principle, its general validity remains a subject of debate. Here, we present evidence that this hypothesis does not hold for a class of mean-field spin glass models. While most of the qualitative physical picture of aging remains unaffected, our results suggest that some important technical aspects should be revisited.



# 1   Introduction

Mean-field spin glasses are prototypes of complex materials. Their rough Hamiltonian function (or "energy landscape") features a multitude of local minima. At finite temperature, this results in a rough free energy landscape, with a multitude of metastable states that can trap the dynamics for times diverging exponentially with system size [1]. In equilibrium, mean-field spin glasses have been solved by several techniques, including the replica and cavity methods, and the structure of thermodynamic states is now well understood [2, 3].

However, such thermodynamic states are by construction inaccessible, because equilibrium cannot be achieved at temperatures below the glass transition. In fact, glassy materials (e.g., window glasses) are usually prepared by cooling from high temperature [4], and the same cooling protocol can also be used to solve optimization problems under the name of simulated annealing [5]. A particular case of such cooling is an instantaneous quench from infinite temperature to zero temperature, i.e. gradient descent dynamics starting from a random initial state. Such dynamics has recently attracted a lot of interest because it is routinely used to train modern deep neural networks [6]. Hence, understanding where, in the rough landscape of a disordered system, a cooling or quench dynamics would end is a problem of primary importance in a broad field of problems, ranging from material science to artificial intelligence [7].

A milestone in this line of research is the exact solution by Cugliandolo and Kurchan of the out-of-equilibrium quench dynamics of the so-called pure spherical $p$-spin-glass model [8]. Here, $p$ refers to the number of interacting spins in the Hamiltonian, and spherical refers to the fact that spins are continuous variables constrained on the $N$-dimensional sphere. These authors were able to solve numerically the exact dynamical mean field theory (DMFT) equations that describe such dynamics when the thermodynamic limit $N \to \infty$ is taken first (at fixed time after the quench). Moreover, they could analytically construct an exact asymptotic solution when time goes to infinity (after the thermodynamic limit) [8, 9]. This solution gave, for the first time, a coherent picture of the low-temperature out-of-equilibrium evolution of disordered systems towards the bottom of their energy landscape, and revealed a series of highly non-trivial physical properties of the dynamics. It shows that (i) the system never becomes stationary but instead *ages* indefinitely, reaching lower and lower regions of the energy landscape; (ii) it asymptotically gets stuck at a "threshold" value of energy that sharply separates high-energy saddle-rich and low-energy minima-rich regions of the landscape [9, 10]; (iii) the threshold level is characterized by "marginal stability", i.e. the spectrum of eigenvalues of the Hessian matrix touches zero, resulting in the presence of arbitrarily soft excitation modes that make the system extremely sensitive to small perturbations; (iv) at long times, any non-linear transformation of time leads to the same result, i.e. the system possesses an internal "clock" that is independent of the actual parametrization of time, the so-called "reparametrization invariance" symmetry [11]; (v) the threshold energy level is asymptotically

sampled uniformly, hence giving rise to a notion of "effective equilibrium" and an associated "effective temperature" [8, 12]; (vi) correspondingly, memory of the initial state is completely lost. This phenomenon of persistent and memoryless aging has been dubbed "weak ergodicity breaking" [9, 13] and has become a central concept in glass physics. In fact, many numerical and experimental studies suggest that structural glasses undergo a similar kind of aging when quenched to low temperatures [4, 14]. Similar results have been obtained for deep neural networks in the under-parametrized regime [15]. The weak ergodicity breaking scenario is particularly attractive because the manifold on which the system evolves asymptotically is independent of the initial condition and can thus be characterized by entirely geometrical methods, without the need for an explicit solution of the dynamics that is difficult to obtain in more complex models [16–21].

Motivated by these considerations, recent work has investigated whether the weak ergodicity breaking scenario, and its asymptotic aging structure, holds more generally in spin glass models. The Ising $p$-spin-glass has been investigated numerically by Rizzo [22] (for $p = 3$) and by Bernaschi et al. [23] (for $p = 2$, corresponding to the Sherrington-Kirkpatrick model). The results of both works suggest either strong ergodicity breaking, i.e. non-vanishing correlation between the initial configuration and that at asymptotically divergent times, or a long-time crossover to a much slower time decay (e.g. logarithmic), thus suggesting that a different asymptotic solution than the Cugliandolo-Kurchan one [24] might apply to these models. Folena et al. [25] studied the mixed spherical $(p + s)$-spin-glass, i.e. a mixture of two pure $p$-spin-glasses, with different values of the number of interacting spins, chosen to be $p = 3$ and $s = 4$. For this $(3+4)$-spin-glass, Folena et al. identified what they called an "onset" temperature $T_{\text{onset}}$, such that for initial configurations prepared in equilibrium at temperature $T > T_{\text{onset}}$, weak ergodicity breaking seemingly applies to gradient descent dynamics, while for $T < T_{\text{onset}}$ one has strong ergodicity breaking [25]. Because of these results, the general validity of the weak ergodicity breaking hypothesis beyond the case of the pure spherical $p$-spin-glass remains undecided.

In this work, we revisit the situation by considering mixed spherical $(p+s)$-spin-glass models with fixed $p = 2$ or $p = 3$, and varying $s$ over a wide range of values. We restrict ourselves to the simplest case of gradient descent (i.e., zero-temperature) dynamics starting from an initial random configuration (i.e., infinite initial temperature). We solve the DMFT equations for these models (hence taking the thermodynamic limit first, at fixed times), both via numerical integration [8] and using series expansions [26]. Our results from both methods consistently suggest that either strong ergodicity breaking holds at any $s > p$, or that weak ergodicity breaking is only restored at very large (unobservable) times via some poorly understood crossover. The phenomenon is most visible at large $s$, but it seems to remain present (although very weakly) even for the $3 + 4$ model investigated by Folena et al. [25].

We show that most of the physical ingredients of the Cugliandolo-Kurchan solution listed above also apply to the mixed $(p + s)$-spin model, namely (i) the system ages indefinitely, (iii) the dynamics approaches a marginally stable manifold, and (v) a modified fluctuation-dissipation relation suggest the emergence of an effective thermal regime. Yet, although our results are not fully conclusive, they strongly suggest that the weak ergodicity breaking hypothesis does not apply, at least on observable time scales, and as a result the system ends up surfing on a non-universal manifold that depends on the initial condition, and whose properties cannot be computed from a simple geometrical scheme. Whether these manifolds can be described by a proper generalization of the Cugliandolo-Kurchan asymptotic solution of DMFT remains an open problem [25].

We note that numerical results on finite-dimensional models of structural glasses seem to agree with the weak ergodicity breaking hypothesis, in the sense that correlation with the initial state is lost at large times, see e.g. [27–29]. While it has been established that in the

infinite-dimensional limit such models are described by a DMFT [30–33], the type of ergodicity breaking in this setting has not been fully investigated, but preliminary results [21] suggest strong ergodicity breaking in infinite dimensions similar to the one we report here for the mixed $(p+s)$-spin glass model. Hence, either weak ergodicity breaking is restored by non-mean-field effects in finite-dimensional glasses, or strong ergodicity breaking is present in these systems but it is too small to be detected numerically. It is important to note that in finite dimensions, under general hypotheses, the asymptotic aging dynamics is connected to the structure of the equilibrium Boltzmann distribution, see Ref. [34] for a review. Progress along these lines would give additional insights on the aging dynamics of glasses and other similarly complex systems.

The rest of this paper is organized as follows. In Sec. 2 we introduce the models we study and review some known properties of their energy landscape and the DMFT equations. In Sec. 3 we present our main results. We conclude by a brief discussion in Sec. 4. A few more technical details are discussed in Appendix.

## 2 Definitions

### 2.1 Models

The Hamiltonian of the pure $p$-spin model is

$$H_p = \sum_{i_1 < i_2 < \ldots < i_p} J_{i_1 i_2 \ldots i_p} \sigma_{i_1} \sigma_{i_2} \cdots \sigma_{i_p}, \tag{1}$$

where the $\sigma_i$ are $N$ real variables satisfying the spherical constraint $\sum_i \sigma_i^2 = N$. The couplings $J_{i_1 i_2 \ldots i_p}$ are i.i.d. Gaussian variables with zero mean and variance $1/(2N^{p-1}p!)$. We consider two classes of mixed $(p+s)$-spin spherical models whose Hamiltonian is a linear combination of two $p$-spin models, each with independent random couplings. The $2+s$ (with $s > 2$) has Hamiltonian

$$H_{2+s}^{\lambda} = \sqrt{\lambda} H_2 + \sqrt{1-\lambda} H_s, \tag{2}$$

where the parameter $0 \leq \lambda \leq 1$ interpolates between the two pure models $H_2$ and $H_s$, and the $3+s$ (with $s > 3$) whose Hamiltonian is

$$H_{3+s}^{\lambda} = \sqrt{\lambda} H_3 + \sqrt{1-\lambda} H_s, \tag{3}$$

again with the interpolating parameter $\lambda$. Following Ref. [35] we define the characteristic polynomial as the covariance between Hamiltonians at different configurations $\sigma, \sigma'$ on the hypersphere

$$f_{p+s}^{\lambda}\left(\frac{\sigma \cdot \sigma'}{N}\right) \equiv N^{-1} \overline{H_{p+s}^{\lambda}(\sigma) H_{p+s}^{\lambda}(\sigma')}, \tag{4}$$

where the overline $\overline{\bullet}$ stands for an average over the $J$s disorder. This is equal to

$$f_{p+s}^{\lambda}(q) = \frac{\lambda q^p + (1-\lambda)q^s}{2}, \quad \text{with} \quad q = \frac{\sigma \cdot \sigma'}{N}, \tag{5}$$

where $p = 2$ or $p = 3$ depending on the considered model, and $q$ is the overlap between two configurations on the hypersphere. The definition and use of the characteristic polynomial $f(q)$ for multi-particle interactions in spin glasses was firstly introduced in Ref. [36].

For each class of models $H_{2+s}^{\lambda}$ and $H_{3+s}^{\lambda}$, we have selected a value of $\lambda$ for each $s$ such as to maximize

$$\Delta E_s^{\lambda} = \frac{f(1)\sqrt{f''(1)}}{f'(1)} - \frac{f'(1) + f(1) + 2}{\sqrt{f''(1)}}. \tag{6}$$

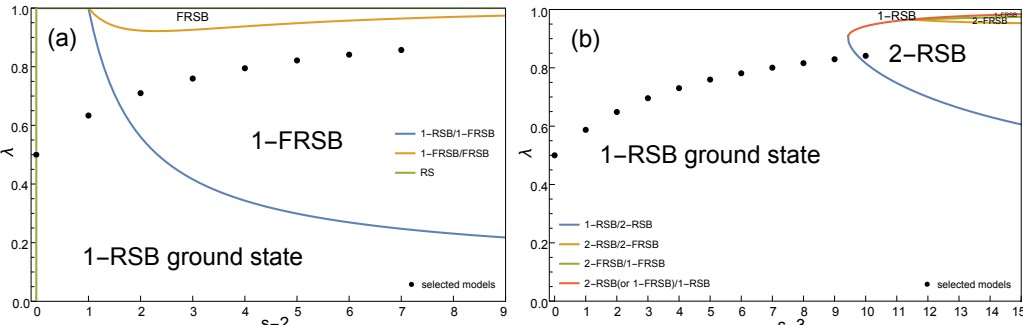

Figure 1: Ground-state of the models studied in this paper. **(a)** Phase diagram for the ground-state of $(2+s)$-spin models for different values of $s$ and $\lambda$. **(b)** Same phase diagram for $(3+s)$-spin models. The plots are derived following Ref. [37].

Table 1: Selected values of $\lambda$ for the $(2+s)$ and $(3+s)$ models.

| | $s$ | 2 | 3 | 4 | 5 | 6 | 7 | 8 | 9 | 10 | 11 | 12 | 13 |
|---|---|---|---|---|---|---|---|---|---|---|---|---|---|
| $2+s$ | $\lambda$ | 0.500 | 0.634 | 0.710 | 0.760 | 0.795 | 0.821 | 0.841 | 0.857 | - | - | - | - |
| $3+s$ | $\lambda$ | - | 0.500 | 0.587 | 0.649 | 0.696 | 0.730 | 0.760 | 0.781 | 0.800 | 0.816 | 0.829 | 0.841 |

Notice that here and in the following, $f'(q) = \partial_q f(q)$. The rationale for this choice of $\lambda$ will be discussed in the following; the selected $\lambda$s are reported in table 1.

## 2.2 Energy landscape

The ground state of the selected models displays different types of replica symmetry breaking (RSB), as shown in Fig. 1. The selected $(2+s)$-spin models present a 1-step RSB (1-RSB) ground-state for $s = 3$ and 1-step+full RSB (1-FRSB) for $s > 3$, with the fullRSB part being located at small values of the overlap [38, 39]. The pure 2-spin ($s = 2$) is peculiar because it presents a trivial landscape with only two minima, and its out-of-equilibrium dynamics is exactly integrable [40]. The selected $(3+s)$-spin models present a 1-RSB ground-state for all $s < 13$, while $s = 13$ presents a 2-RSB ground-state [37].

Above the ground state, each of the selected models presents a rough energy landscape, with an exponential (in the system size) number of local minima. In order to characterize this complex landscape we define three different energies:

- $E_{gs}$, the ground-state energy, is the lowest energy of the landscape [39, 41].

- $E_{th}$, the threshold energy, is the energy above which typical stationary points are saddles, and below which they are minima [8, 25, 42–44].

- $E_{alg}$, the algorithmic energy, is the lowest energy reachable by an optimization algorithm which computes the gradient of the Hamiltonian a finite number of times [45–48].

Each of these energies can be exactly evaluated for arbitrary mixed models. Below we only report calculations for both $E_{gs}$ and $E_{th}$ in the simplest case of a 1-RSB landscape, but the expressions can be generalized to any level of RSB. We define the complexity as the average over the disorder of the logarithm of the number $\mathcal{N}$ of stationary points with given energy (per spin) $E_{IS}$. This is evaluated by the Kac-Rice formula

$$\Sigma(E_{IS}) \equiv N^{-1}\overline{\log\left(\mathcal{N}(E_{IS})\right)}, \quad \text{with} \quad \mathcal{N}(E_{IS}) = \int_{\sigma \in \mathcal{S}^N} d\sigma \, \delta(H - NE_{IS}) \, \delta(\nabla H) \, |\det(\nabla^2 H)|, \tag{7}$$

with $\int_{\sigma \in \mathcal{S}^N} d\sigma$ the integral over the space of configurations (hypersphere).

Assuming a 1-RSB ansatz, the complexity of the energy landscape in mixed $p$-spin models is [25, 49, 50]

$$\Sigma^{(1)}(\chi) = \frac{1}{2}\left( \chi^2 f'(1) + \log\left( \frac{1}{\chi^2 f'(1)} \right) - f(1)\left( \frac{1}{\chi f'(1)} - \chi \right)^2 - 1 \right), \qquad (8)$$

where $\chi$ is the linear susceptibility associated to typical local minima with energy

$$E_{\text{IS}}^{(1)}(\chi) = -f(1)\left( \frac{1}{\chi f'(1)} - \chi \right) - \chi f'(1). \qquad (9)$$

The superscript $^{(1)}$ reminds that these expressions hold under the 1-RSB ansatz, and the subscript $_{\text{IS}}$ stands for inherent structure, the name usually given to local minima in the glass literature. A parametric plot of $\Sigma$ versus $E$, eliminating $\chi$, gives the complexity of local minima as a function of the energy level in the landscape. If the ground-state, or some of the higher-energy states, present a more complex RSB structure, then Eqs. (8) and (9) are not exact, and we must resort to more involved calculations, see Ref. [39] for 1-FRSB and Ref. [44] for 2-RSB.

The ground-state energy corresponds to the energy at which the complexity vanishes, therefore within the 1-RSB ansatz

$$E_{gs}^{(1)} = E_{\text{IS}}^{(1)}(\chi_{gs}^{(1)}), \quad \text{with} \quad \chi_{gs}^{(1)} \quad \text{s.t.} \quad \Sigma(\chi_{gs}^{(1)}) = 0. \qquad (10)$$

The energy $E_{th}$ is defined as the energy at which dominant minima become saddles, i.e. the vibrational spectrum is marginal. The vibrational spectrum (see e.g. Ref. [35]) follows a semicircular law $\rho_\mu(\lambda)$ of radius $R = 2\sqrt{f''(1)}$ centered at $\mu$, where $\mu$ is given in terms of the susceptibility by inverting

$$\chi(\mu) = \int d\lambda \frac{\rho_\mu(\lambda)}{\lambda} = \frac{\mu - \sqrt{\mu^2 - \mu_{mg}^2}}{2f''(1)}, \qquad (11)$$

with the marginal value thus corresponding to $\mu_{mg} = R = 2\sqrt{f''(1)}$. Note that the results on the spectrum hold for any level of RSB. The corresponding energy (at 1-RSB level) and susceptibility are

$$E_{th}^{(1)} = E_{IS}^{(1)}(\chi_{mg}), \quad \text{with} \quad \chi_{mg} = \frac{1}{\sqrt{f''(1)}}. \qquad (12)$$

The algorithmic energy $E_{alg}$ is the minimal energy reachable by a search algorithm running in polynomial time in system size. This energy can be reached by moving in the Thouless-Anderson-Palmer (TAP) free energy landscape from magnetization $\langle \sigma_i \rangle = m_i = 0$ (center of sphere), in $N$ orthogonal unit steps until reaching the surface of the sphere defined by $\sum_i m_i^2 = N$. In physical terms, this is a sort of annealing in temperature with a re-weighting of the Hamiltonian, see Ref. [45, 48]. The algorithmic energy reads

$$E_{alg} = \int_0^1 dq f''(q)^{1/2}, \qquad (13)$$

and the corresponding ansatz is intrinsically of the continuous fullRSB kind. We notice that in the case of a continuous fullRSB ground state, one has $E_{gs} = E_{th} = E_{alg}$ [45], i.e. the ground state energy (up to subleading corrections in $1/N$) can be reached in polynomial time.

Finally, we can define a value of energy at which the 1-RSB complexity has a maximum,

$$E_{max}^{(1)} = E_{IS}^{(1)}(\chi_{max}^{(1)}), \quad \text{where} \quad \chi_{max}^{(1)} = \frac{\sqrt{f(1)}}{\sqrt{f'(1)^2 - f(1)f'(1)}}, \quad s.t. \quad \partial_\chi \Sigma^{(1)}(\chi) = 0. \qquad (14)$$

This value of energy is located above $E_{th}^{(1)}$ and has no particular physical meaning, because the complexity is ill-defined and unstable towards further RSB in that region [51,52]. Yet, we take the quantity $\Delta E_s^\lambda = E_{max}^{(1)} - E_{th}^{(1)}$ reported in Eq. (6) as an estimate of the energy range in which "non-trivial" effects may take place in the landscape, and this is why we choose $\lambda$ such as to maximize $\Delta E_s^\lambda$. We stress once again that this choice has no obvious physical meaning and is just one among several possible ways to choose a value of $\lambda$ for each $s$, which is helpful to reduce the parameter space of the model.

## 2.3 Gradient descent dynamics

In this work we consider the simplest form of local greedy dynamics, the gradient descent (GD) dynamics defined by

$$\partial_t \sigma_i = -\mu \sigma_i - \nabla H_i, \qquad \mu = \nabla H(t) \cdot \sigma(t)/N, \tag{15}$$

where the term $\mu(t)$, also called "radial reaction", is added in order to enforce the spherical constraint. The system is prepared in an initial random configuration (on the sphere) and the gradient of the Hamiltonian is then followed until reaching a local minimum. In the $N \to \infty$ limit, for any mixed model of covariance $f(q)$, the GD dynamics can be rewritten in terms of correlation $C_{tt'} = \langle \sum_{i=1}^N s_i(t)s_i(t')\rangle/N$ and response $R_{tt'} = \langle \sum_{i=1}^N \delta s_i(t)/\delta h_i(t')\rangle/N$, being $\langle\rangle$ the average over different random initial conditions and different quenched disorder of the couplings $J$s. An external field $h_i$ is added to the GD equations to compute the linear response [8]. The resulting DMFT equations for the GD dynamics read [8]

$$
\begin{aligned}
\partial_t C_{tt'} =& -\mu_t C_{tt'} + \int_0^t ds f''(C_{ts})R_{ts}C_{st'} + \int_0^{t'} ds f'(C_{ts})R_{t's}, \\
\partial_t R_{tt'} =& \delta_{tt'} - \mu_t R_{tt'} + \int_{t'}^t ds f''(C_{ts})R_{ts}R_{st'}, \\
\mu_t =& \langle \nabla H(t) \cdot \sigma(t)\rangle/N = \int_0^t \left( f''(C_{ts})R_{ts}C_{ts}ds + f'(C_{ts})R_{ts}\right)ds,
\end{aligned}
\tag{16}
$$

where $\mu_t$ is the average Lagrange multiplier enforcing the spherical constraint. The energy is given by

$$E_t = \langle H(t)\rangle = -\int_0^t ds f'(C_{ts})R_{ts}. \tag{17}$$

We notice that these equations do not present any explicit dependence on the starting configuration, while a term proportional to $C_{t0}$ is found if the dynamics is initialized in equilibrium at finite temperature [25]. For a broader and pedagogical introduction to GD in mean-field spin-glass models, see e.g. Ref. [7,53,54].

These equations were first studied in the out-of-equilibrium setting in Ref. [8] (for an arbitrary temperature of the thermal bath). There, an ansatz for the long-time dynamics was proposed, resulting in an asymptotic energy that coincides with the 1-RSB threshold energy $E_{th}^{(1)}$ that separates minima from saddles. These studies were carried out on the pure $p$-spin model, which has the special property that all the marginal minima have exactly energy $E_{th}^{(1)}$. In Refs. [25,35], the situation was shown to be different for the more general mixed (3+4)-spin model, for which marginal minima can be found in a broad range of energies. Yet, Refs. [25,35] concluded that for this $(3 + 4)$-spin model, GD initialized in random configurations would converge to the asymptotic solution of Ref. [8] and thus to energy $E_{th}^{(1)}$.

In more general models, it is known that $E_{th}^{(1)}$ can go below the ground state energy. Because the correctness of Eqs. (16) has been mathematically proven [55], the ansatz of Ref. [8]

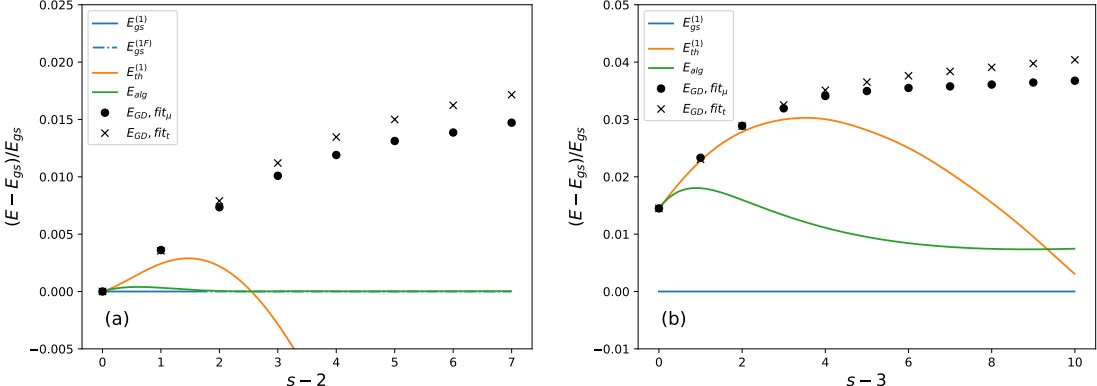

Figure 2: Asymptotic energy reached by the gradient descent (GD) dynamics from random initial condition, compared to the ground state energy $E_{gs}$, the threshold energy $E_{th}^{(1)}$ and the algorithmic energy $E_{alg}$. Black dots and crosses indicates two different extrapolations of the gradient descent dynamics, respectively with radial reaction $\mu$ and with time as abscissa. **(a)** Asymptotic energies reached by $(2+s)$-spin models. For $E_{gs}$ the continuous line is the 1-RSB solution while the dashed-dotted is the 1-FRSB solution (barely distinguishable from $E_{alg}$). **(b)** Asymptotic energies reached by $(3+s)$-spin models. Note that the $(3+13)$ model has a 2-RSB ground state, but here we reported the 1-RSB result (see [44] for further details). While for $s = p$ (and, within numerical precision, for $s \gtrsim p$) gradient descent reaches $E_{th}$, for $s > p$ the gap between the two energies increases quickly.

cannot apply in such cases and must be generalized. In this work, we thus revisited the results of Refs. [25, 35] by numerically integrating Eqs. (16) in a similar way as done before, but considering a broader range of values of $p$ and $s$ in the mixed $(p+s)$-spin model. Furthermore, we also considered a series expansion of the equations as in Ref. [26]; the details of this method can be found in appendix B.

## 3 Results

Starting from a random configuration and performing a GD dynamics, we observe that the energy asymptotically reaches values above the threshold energy $E_{th}^{(1)}$ predicted by the asymptotic solution derived in Ref. [8] (Fig. 2). Yet, it remains true that the asymptotically reached local minima are marginal, i.e. their spectrum has almost flat directions. This is not in contradiction with the structure of the energy landscape, which presents a large number of marginal minima (exponential in $N$) even above the threshold energy [25].

The main claim of Ref. [25] is that preparing a system in equilibrium at a finite temperature (below $T_{\text{onset}}$) and then running GD dynamics, it will asymptotically reach marginal minima with energies below $E_{th}^{(1)}$, aging in a confined space. However, starting from random initial conditions (or preparing above the onset temperature $T_{\text{onset}}$) the system reaches the threshold energy $E_{th}^{(1)}$. Here we claim instead that preparing the system in a random configuration, the GD dynamics reaches energies above $E_{th}^{(1)}$. The fact that such behaviour was not observed in Ref. [25] is because in mixed $(p + s)$-models with close $p$ and $s$ the effect is very small, as can be seen in Fig. 2.[1] Furthermore, contrarily to what was proposed in Ref. [25], we propose that

---

[1]Notice that the $(3 + 4)$-spin model of Ref. [25] corresponds to $\lambda = 0.5$ (here instead we fix $\lambda = 0.587$ for the

the long-time limit of the correlation $C_{t0}$ between the initial and final configurations does not vanish; again, for close $p$ and $s$ the effect is so small that it went undetected in previous work. In the scenario we propose here, initializing the GD dynamics in equilibrium at temperature $T$ would always result in a $T$-dependent asymptotic energy, with $E_{th}^{(1)}$ playing no special role, and no sharp $T_{onset}$ would then be defined.

The results presented in this section are obtained by numerically integrating the DMFT Eqs. (16) with a fixed time step $dt$ as detailed in [25]. We have chosen a time step $dt = 0.05$, which approximate 'well' the long time dynamics for all considered models (see appendix A). In order to support our numerical findings (obtained via integration of the DMFT), we have solved the same equations by using a series expansion in the two times $t, t'$ as was first suggested in Ref. [26]. The obtained results confirm the findings from the numerical integration. The two methods, the "integration" and the "series expansion", give different insight on the problem. The first allows for a longer time span (up to 1500 time units, with time step 0.05), while the second (that can span up to 100 time units) allows one to precisely evaluate derivatives on the spanned region, which are needed to precisely evaluate power-law decays.

We would like to stress, however, that both methods give access to a limited time interval, and in absence of an analytic solution, the infinite-time limit remains inaccessible. The possibility that the scenario we propose is only a pre-asymptotic regime that would crossover to a weak ergodicity regime thus remains open.

## 3.1 Power-law decay with series expansion

Focusing on the GD dynamics starting from a random initial configuration, three independent observables are considered: energy $E(t)$, radial reaction $\mu(t)$ and overlap with the initial condition $C(t, 0) = C_{t0}$. In the long time limit (in the aging regime), according to the asymptotic analysis of Ref. [8], we expect them to asymptotically follow three independent power laws:

$$\Delta E(t) \sim t^{-\alpha_E}, \qquad \Delta \mu(t) \sim t^{-\alpha_\mu}, \qquad \Delta C(t, 0) \sim t^{-\alpha_C}, \tag{18}$$

where $\Delta O(t) = O(t) - \lim_{t\to\infty} O(t)$ for each observable. In the case of a quench to the critical temperature (for a fullRSB transition), exact relations between the $\alpha_E$ and $\alpha_C$ exponents were found in Ref. [56].

In order to numerically study the power laws describing the asymptotic dynamics, we adopt the idea –firstly introduced in Ref. [26]– of expanding the integro-differential DMFT equations describing the out-of-equilibrium dynamics in powers of the two times $t, t'$. The obtained series has a small radius of convergence (of order one). A Padé approximation is thus performed in order to extract useful information for the long-time dynamics, which consists in rewriting polynomials of degree $L$ in terms of fractions of polynomials of degree $L/2$, such that they have the same Taylor expansion. The attempt in Ref. [26] was conditioned by two important limitations, the computing power and a probable floating-point approximation error in the evaluated coefficients that has biased the Padé approximations and therefore the asymptotic results. In order to overcome this second difficulty we use a multiple-precision floating-point library (GNU MPFR) that allows one to keep an arbitrarily large number of digits for every coefficient in the expansion, thus avoiding numerical errors in the resummation. The only limitation is then due to the number of terms that can be computed (between 1000 and 2000 coefficients), which when resummed with the Padé approximation, gives access to times $\sim 100$, to be compared to the times $\sim 1000$ reached by a simple integration algorithm. The advantage of the series expansion is that we can also evaluate derivatives without suffering

---

same model), which results in an even smaller discrepancy between $E_{th}^{(1)}$ and the extrapolated $E_{GD}$ than the one shown in Fig. 2. More generally, the values of $\lambda$ that we choose in this paper "almost" maximize the discrepancy for each given mixed $(p + s)$ model.

Table 2: Power-law exponents in the $(2+s)$ and $(3+s)$ models. The exact results for the 2-spin model are shown in parenthesis (see Ref. [40]). Values have errors between 0.001 and 0.05 as shown in the error bars of Fig. 3, estimated by comparing different fitting ranges.

| | $s$ | 2 | 3 | 4 | 5 | 6 | 7 | 8 | 9 | 10 | 11 | 12 | 13 |
|---|---|---|---|---|---|---|---|---|---|---|---|---|---|
| | $\alpha_E$ | 0.999 (1) | 0.681 | 0.604 | 0.547 | 0.508 | 0.481 | 0.462 | 0.447 | - | - | - | - |
| $2+s$ | $\alpha_\mu$ | 0.999 (1) | 0.702 | 0.678 | 0.663 | 0.652 | 0.636 | 0.647 | 0.632 | - | - | - | - |
| | $\alpha_C$ | 0.748 (3/4) | 0.506 | 0.464 | 0.424 | 0.407 | 0.401 | 0.380 | 0.369 | - | - | - | - |
| | $\alpha_E$ | - | 0.666 | 0.657 | 0.617 | 0.566 | 0.538 | 0.503 | 0.474 | 0.453 | 0.436 | 0.423 | 0.411 |
| $3+s$ | $\alpha_\mu$ | - | 0.666 | 0.670 | 0.663 | 0.655 | 0.650 | 0.643 | 0.637 | 0.631 | 0.624 | 0.618 | 0.615 |
| | $\alpha_C$ | - | 0.375 | 0.338 | 0.313 | 0.299 | 0.285 | 0.283 | 0.277 | 0.284 | 0.266 | 0.271 | 0.266 |

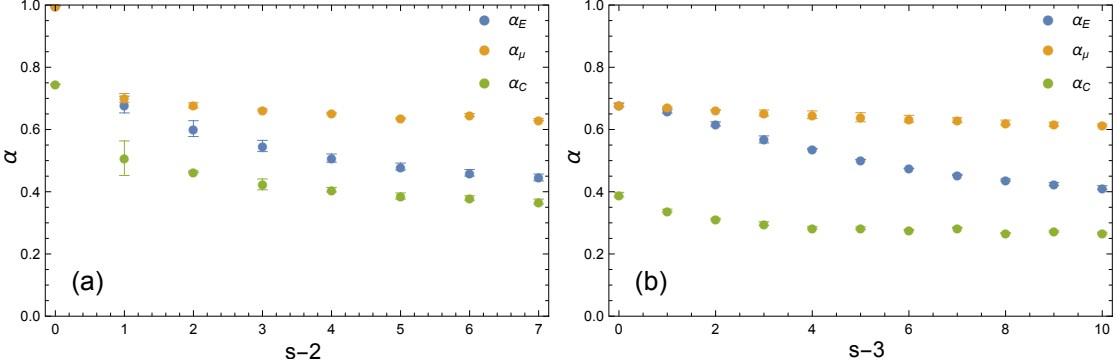

Figure 3: Exponents $\alpha_E, \alpha_\mu, \alpha_C$ extrapolated from the series expansion of the DMFT equations (see appendix B and table 2). These describe the decay of the energy $E(t)$, the radial reaction $\mu(t)$ and the correlation with the initial condition. **(a)** Power-law exponents in $(2+s)$-spin models. **(b)** Power-law exponents in $(3+s)$-spin models.

from discretization errors due to integration. In appendix B we explain in details the procedure we used to extract the exponents $\alpha_E, \alpha_\mu, \alpha_C$ from the series expansion. In a nutshell, we extract the exponents from the asymptotic limit of the ratio $t\partial_t^2 O(t)/\partial_t O(t)$, $O(t)$ being either $E(t)$, $\mu(t)$ or $C(t,0)$.

The results are shown in Fig. 3 and numerical values are given in table 2. The error bars are based on a fitting procedure over a Padé series (as reported in appendix B), and therefore must be considered as attempted estimation of the errors. Only in pure models $\alpha_E = \alpha_\mu$, because the energy is proportional to the radial reaction [35]. For the special case of the pure 2-spin the fitted power law agrees with the analytically known one [40], i.e. $\alpha_E = \alpha_\mu = 1$ and $\alpha_C = 3/4$. For the pure 3-spin we conjecture that the exact values are $\alpha_E = \alpha_\mu = 2/3$ and $\alpha_C = 3/8$. These coefficients are not universal, and it remains an open question whether there exists some equations relating them, in the spirit of Ref. [56].

## 3.2 Is the asymptotic dynamics marginal?

In Fig. 4 we show the time evolution of the radial reaction $\mu(t)$. We confirm that the gradient descent dynamics from random configurations asymptotically approaches configurations that have a marginal spectrum [8], i.e. the radial reaction approaches asymptotically its marginal value $\mu_{mg} \equiv 2\sqrt{f''(1)}$ [25, 35, 49]. The convergence of $\mu(t)$ towards $\mu_{mg}$ is controlled by different power-law decays depending on the model. In the inset of Fig. 4 we show that the exponents $\alpha_\mu$ derived from the series expansion can be used to confirm the convergence towards the marginal value: indeed, the curves appear perfectly linear when plotted as a

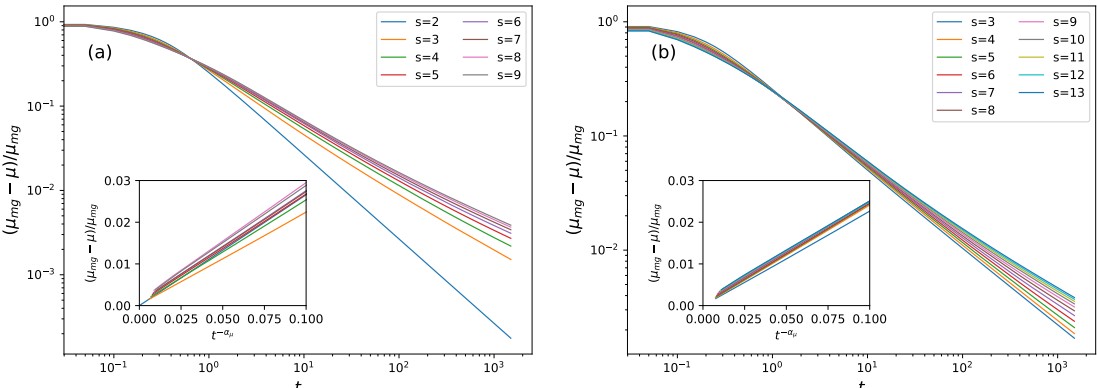

Figure 4: Reduced radial reaction $(\mu_{mg} - \mu(t))/\mu_{mg}$ vs time $t$. It is asymptotically expected to reach zero if the final configuration is marginal. In the inset, the time axis is rescaled as $t^{-\alpha_\mu}$, to show a linear decay. Notice that $\alpha_\mu$ is evaluated without any assumption on the marginality, i.e. without assuming $\lim_{t \to \infty} \mu(t) = \mu_{mg}$, as explained in appendix B. **(a)** Time dependence of the rescaled radial reaction in $(2 + s)$-spin models. **(b)** Time dependence of the rescaled radial reaction in $(3 + s)$-spin models.

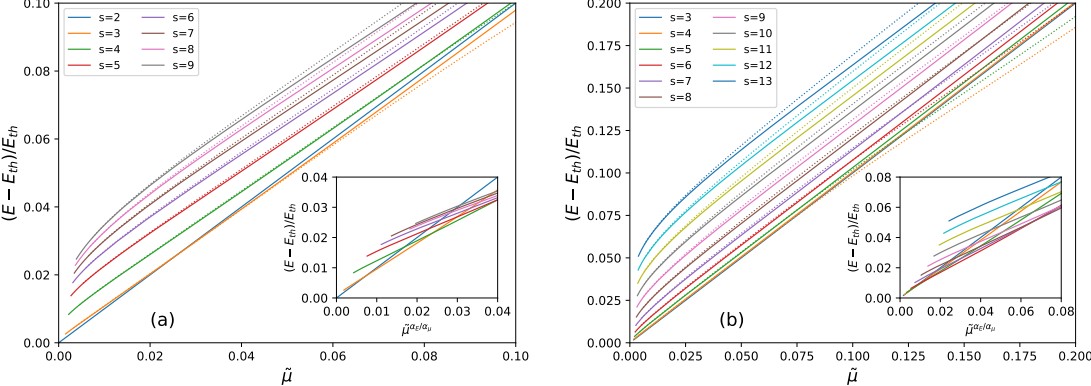

Figure 5: Reduced energy vs reduced radial reaction. The inset shows the same data rescaled with the power-law exponent $\alpha_E/\alpha_\mu$ in order to obtain a linear behavior, which once fitted gives the estimates of the asymtotic energy shown in Fig. 2 (black dots). Dotted lines shows the results for the same system with a different local dynamics, as described in section 3.5. **(a)** Results for $(2 + s)$-spin models. For $s > 3$ the threshold is evaluated with the 1-FRSB ansatz [39]. **(b)** Results for $(3 + s)$-spin models, all the thresholds are evaluated with the 1-RSB ansatz.

function of $t^{-\alpha_\mu}$, and the linear extrapolation to infinite times coincides with $\mu_{mg}$. We notice that the evaluation from series expansion of $\alpha_\mu$ (for each model) does not assume $\mu = \mu_{mg}$, therefore the linear convergence towards zero (in the inset of Fig. 4) is a strong confirmation of the system being asymptotically marginal.

## 3.3 What is the asymptotic energy?

Having established that the gradient descent dynamics is asymptotically marginal for every model considered, we can employ the reduced radial reaction $\tilde{\mu}(t) = (\mu_{mg} - \mu(t))/\mu_{mg}$ as a measure of time. It is $\tilde{\mu}(0) = 1$ at the initial time and reaches $\tilde{\mu}(\infty) = 0$ asymptotically. We

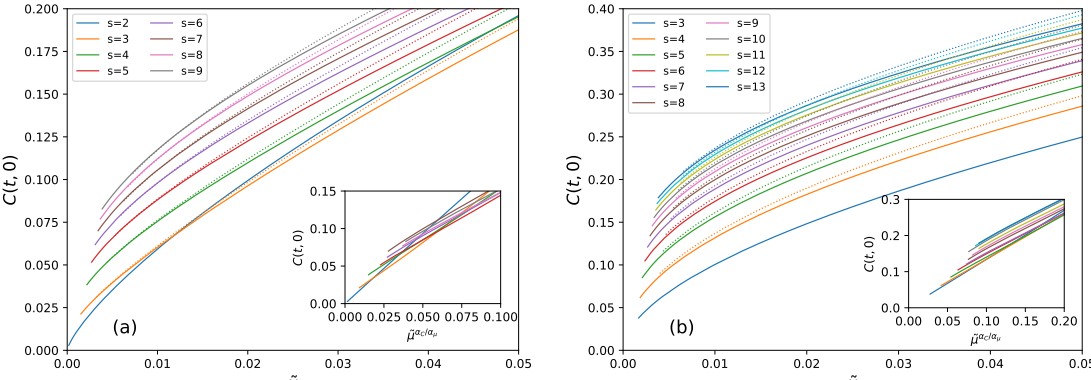

Figure 6: Correlation with the initial condition vs reduced radial reaction. The insets shows the same data rescaled with the power-law exponent $\alpha_C/\alpha_\mu$ in order to display a linear behaviour. The dotted lines correspond to the same system with a persistent short-time dynamics, as described in section 3.5. For $s > p$, the results suggest an asymptotic memory of the initial configuration, consistent with a strong ergodicity breaking scenario. **(a)** Results for $(2 + s)$-spin models. **(b)** Results for $(3 + s)$-spin models.

can then study the energy decay as a function of $\tilde{\mu}$, as shown in Fig. 5. Increasing $s$ for both $(2 + s)$ and $(3 + s)$ models has the effect to increase the disagreement with the hypothesis of convergence to the threshold $E_{th}$, both in the case of a 1-RSB and 1-FRSB ansatz, the second being used for $s > 3$ in the $(2+s)$ model. The insets of Fig. 5 show the same data as a function of $\tilde{\mu}$ rescaled with the power-law exponents deduced from the series expansion. The resulting linear behavior is fitted to obtain the asymptotic estimates shown in Fig. 2 (black dots). A similar analysis has been done using the rescaled time $t^{-\alpha_E}$ to obtain a slightly different fit, also shown in Fig. 2 (black crosses). Despite not being exactly coincident, the estimates of the asymptotic energy from the two fits suggest that the energy reaches values that are well above the threshold.

## 3.4 Is there a strong ergodicity breaking?

One very important question is whether the aging dynamics can decorrelate from any previously reached configuration over a long enough time. This is referred to as the weak ergodicity breaking ansatz, and it is one of the main ingredients that allowed the derivation of the asymptotic solution of the DMFT equations of the pure $p$-spin model [8,9]. Recently, in Refs. [22,23,25], it has been suggested that weak ergodicity breaking could be non-universal, and some systems could age in a confined part of the phase space, thus showing strong ergodicity breaking, i.e. a persistent correlation with the initial condition. Our results for the GD dynamics of mixed $(p + s)$-spin models starting from a random configuration seem to confirm strong ergodicity breaking. In Fig. 6 we show the correlation with the initial configuration $C(t, 0)$ as a function of the reduced $\tilde{\mu}$. The inset shows the same data rescaled with the power-law exponent $\alpha_C/\alpha_\mu$ derived from the series expansion. The linearized curves suggest a non-zero asymptotic value for the correlation, i.e. $\lim_{t\to\infty} C(t, 0) \neq 0$, at least if we assume that the time of integration ($t = 1500$) is long enough to observe the asymptotic behavior. In order to further support this observation we have looked at the two-time correlation $C(t, t_w) = C_{tt_w}$ for different waiting times $t_w$. The results are shown in Fig. 7, with as abscissa the radial reaction "time difference" $(1/\tilde{\mu}(t) - 1/\tilde{\mu}(t_w))^{-1}$. As observed in Ref. [23], for increasing waiting time $t_w$ the system decorrelates less and less, which once again suggests

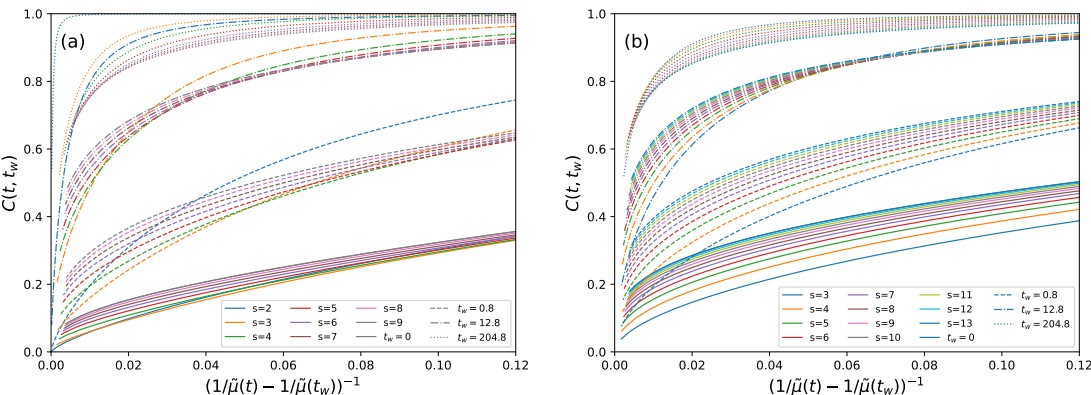

Figure 7: Two-time correlations $C(t, t_w)$ vs $(1/\tilde{\mu}(t) - 1/\tilde{\mu}(t_w))^{-1}$ for different waiting times $t_w = 0, 0.8, 12.8, 204.8$. For $s > p$, the data suggest a strong ergodicity breaking scenario, i.e. $\lim_{t \to \infty} C(t, t_w) \neq 0$, for all $t_w$. **(a)** Results for $(2 + s)$-spin models. **(b)** Results for $(3 + s)$-spin models.

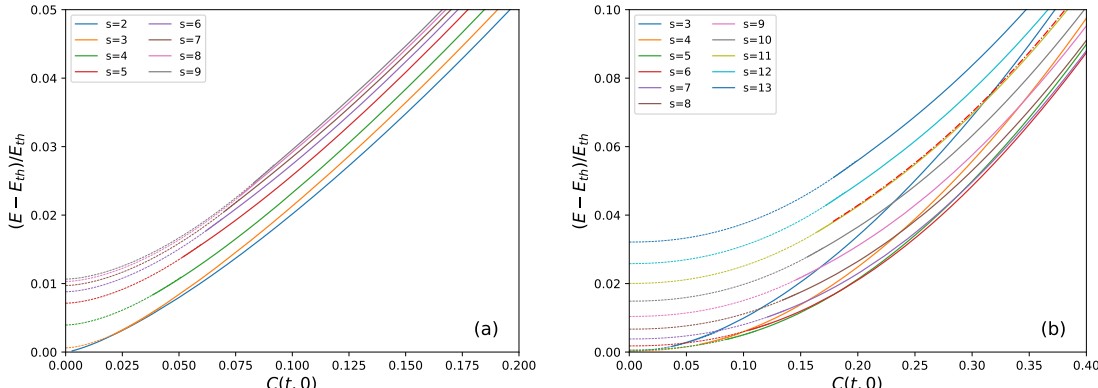

Figure 8: Reduced energy vs correlation with the initial condition. A power-law fit (dashed lines) is performed on the last 500 points of each curves. The fits suggest that even in the limit $C(t, 0) \to 0$ the asymptotic energy would be above the threshold one. **(a)** Results for $(2 + s)$-spin models. **(b)** Results for $(3 + s)$-spin models. The red dashed-dotted line shows the extrapolated ($dt \to 0$) curve for $s = 11$. In appendix A we discuss the $dt$-scaling of the error due to the discretization.

strong ergodicity breaking. Furthermore, notice that even assuming that the dynamics eventually become uncorrelated with the initial condition, the asymptotic behavior of the parametric curve $E(t)$ versus $C(t, 0)$ indicates an asymptotic energy above the threshold value (Fig. 8).

## 3.5 Are the results robust against a change of the short-time dynamics?

In this section we investigate to what extent the asymptotic dynamics is influenced by changes in the short-time dynamics. We modify the thermal bath by adding an exponential persistence, i.e. the time-derivative operator is changed into

$$\partial_t C \longrightarrow \left(\partial_t + \int_{-\infty}^{t} ds K_{ts} \partial_s\right) C, \quad \text{where} \quad K_{tt'} = \gamma \exp(-|t - t'|/\tau). \tag{19}$$

At finite temperature, in order to preserve the detailed balance condition, the thermal noise is changed accordingly as

$$\langle \xi(t)\xi(t') \rangle = 2T\delta(t-t') \longrightarrow 2T\Big(\delta(t-t')+K_{tt'}\Big), \tag{20}$$

but since we work here at $T = 0$, this change is irrelevant. The parameters $\tau$ and $\gamma$ can be tuned to define the strength of the persistence. The resulting equations for correlation and response are

$$
\begin{aligned}
\partial_t C_{tt'} = & -\mu_t C_{tt'} + \int_0^t ds\Big(f''(C_{ts})R_{ts}-K_{ts}\partial_s\Big)C_{st'} + \int_0^{t'} ds\Big(f'(C_{ts})-TK_{ts}\Big)R_{t's}, \\
\partial_t R_{tt'} = & \delta_{tt'} - \mu_t R_{tt'} + \int_{t'}^t ds\Big(f''(C_{ts})R_{ts}-K_{ts}\partial_s\Big)R_{st'},
\end{aligned}
\tag{21}
$$

to be compared with Eqs. (69) and (71) of Ref. [57]. We have integrated and expanded in series the Eqs. (21) for $\gamma = 1$ and $\tau = 1$. We found that the exponential persistence gives just a linear rescaling of the characteristic time (for $t > \tau$), i.e. given a one-time observable $O(t)$ of the original GD dynamics, its analog $O^{pers}(t)$ in the persistent dynamics has an asymptotic behaviour $O^{pers}(t) = O(t/t_{pers})$ with $t_{pers} > 1$ (that depends on the specific model). This behavior is confirmed by looking at parametric plots where the time has been substituted by the reduced radial reaction $\tilde{\mu}$. In Fig. 5 and Fig. 6 the dotted lines corresponds to the persistent GD dynamics and are asymptotically matching the original GD dynamics (continous lines).

We conclude that a modification of the short-time dynamics does not seem to have any impact on the asymptotic behavior, suggesting that a possible closure of the asymptotic DMFT equations is still possible, despite the lack of weak ergodicity breaking. In other words, while the dynamics remains confined in a region of space that depends on the initial condition, such region is asymptotically explored in a way that does not depend on short-time details, suggesting some kind of effective thermal behavior. In this regard, in order to completely overcome the short-time dynamics, a possible solution would be to study the quasi-equilibrium dynamics introduced in Ref. [58–61].

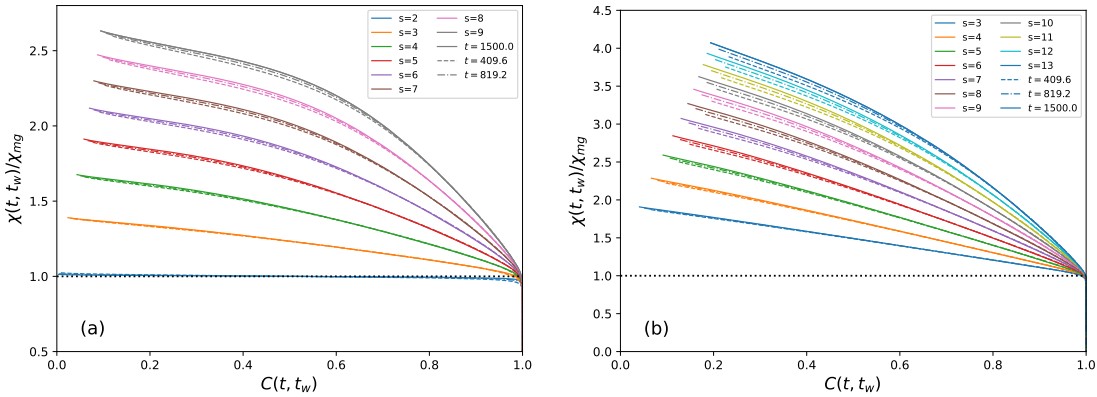

Figure 9: Parametric plot over waiting time $t_w$ of the rescaled integrated response $\chi(t,t_w)/\chi_{mg}$ vs the correlation $C(t,t_w)$, for several fixed values of time $t = 25.6, 204.8, 1500$. **(a)** Results for $(2+s)$-spin models. **(b)** Results for $(3+s)$-spin models. For both classes of models the linear 1-RSB ansatz is not able to describe the asymptotic behaviour (see Fig. 10 for a detailed comparison). The dynamics seems to asymptotically reach a different unknown ansatz.

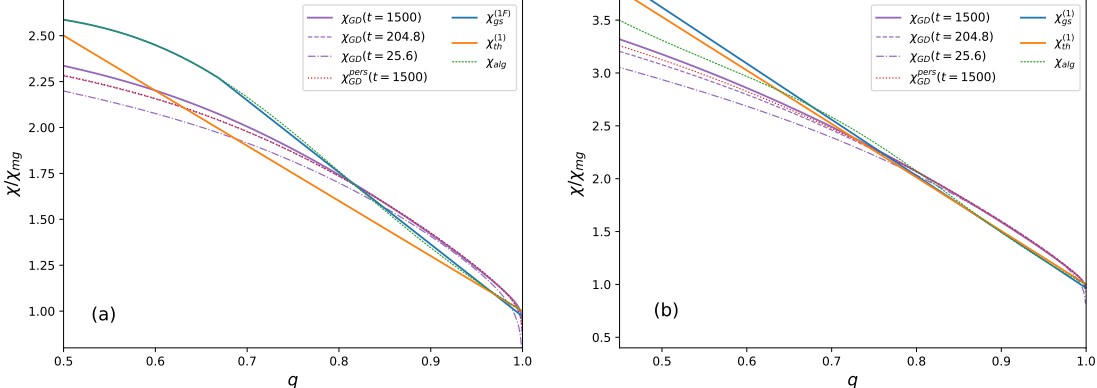

Figure 10: Parametric plot over waiting time $t_w$ of the rescaled integrated response $\chi(t, t_w)/\chi_{mg}$ vs the correlation $C(t, t_w)$, for several fixed values of time $t = 25.6, 204.8, 1500$, compared with several known asymptotic references: static ($\chi_{gs}$), dynamical 1-RSB ($\chi_{th}^{(1)}$) and algorithmic ($\chi_{alg}$) ansatz. **(a)** $(2+9)$-spin model. **(b)** $(3+12)$-spin model. The GD dynamics with persistence (red dotted line), shows the same asymptotic FDR shape. For $q$ close to 1 the FDR is concave, which is incompatible with any known thermodynamic calculation (see Appendix C).

## 3.6 Is there a well defined effective temperature?

A standard way to analyze the aging dynamics [8] is by looking at the parametric plot of integrated response $\chi(t, t_w) = \int_{t_w}^{t} ds R(t, s)$ versus the correlation $C(t, t_w)$. These are shown in Fig. 9 for different models. One can introduce an effective temperature that quantifies the violation of the equilibrium fluctuation-dissipation relation (FDR), and is given by

$$x^t[C(t, t_w)] = -\frac{d\chi(t, t_w)}{dC(t, t_w)}. \tag{22}$$

In pure $p$-spin models the 1-RSB ansatz for the GD dynamics gives, at long times, a unique effective temperature independent on the correlation, $x = \frac{1}{\chi_{mg}f'(1)} - \chi_{mg}$, which is also equal to the derivative of the complexity at the threshold energy $E_{th}$ [8]. In Fig. 9b we see that general mixed $(3+s)$-spin models do not exhibit a unique effective temperature (as given by a 1-RSB anstaz) but rather a varying temperature $x^t[q]$ that depends on the correlation $q = C(t, t_w)$. The same can be said for the FDR of $(2+s)$ models. Moreover, as shown in Appendix C, even more refined "thermodynamic solutions" to the asymptotic behavior (such as a 1-FRSB dynamical ansatz [39]), do not agree with dynamical observations. In other words, the GD dynamical overlap probability $P_{GD}^t(q) = x'_{GD} = -\chi''_{GD}$ does not converge to any expected replica ansatz.

This is not only true because the support of $P_t^d(q)$ does not reach $q = 0$ (due to strong ergodicity breaking), but also because of the actual shape of the solution. This is shown in more details in Fig. 10 where $\chi_{GD}(q)$ for different times $t = 25.6, 204.8, 1500$ are compared with the static ground state ($\chi_{gs}$), threshold ($\chi_{th}$) and optimal ansatz ($\chi_{alg}$). The most clear evidence that $P_{GD}^t(q)$ is not converging to the known $\chi_{th}$ ansatz is given by $\chi_{GD}(q)$ for $0.8 < q < 1$, for which the curves for different times $t = 25.6, 204.8, 1500$ seem to have converged to a concave (not 1-RSB nor 1-FRSB) solution. This gives further evidence that our actual understanding of the asymptotic dynamics is far from being complete. We note that similar results are obtained by looking at the fluctuation-dissipation ratio for the persistent GD dynamics (red dotted line) in Fig. 10, which supports the claim that the asymptotic limit of $P_{GD}^t(q)$ is also independent of the short-time details of the dynamics.

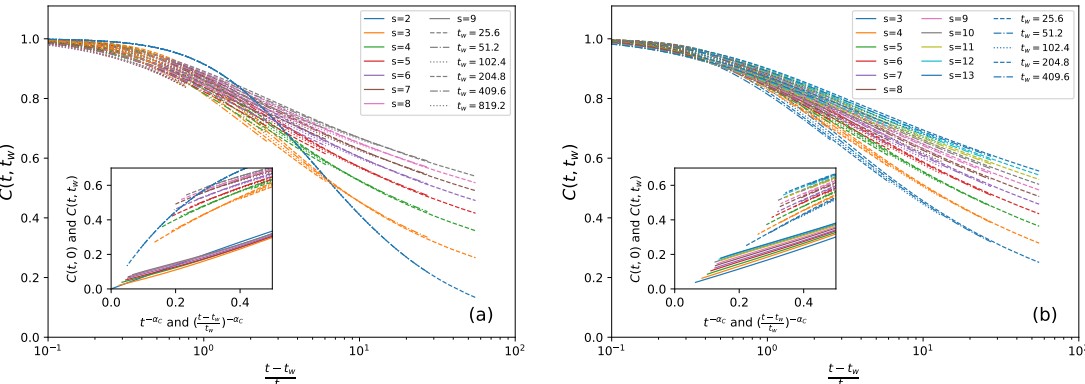

Figure 11: Correlation vs rescaled time $(t - t_w)/t_w$ for different waiting times $t_w = 25.6, 51.2, 102.4, 204.8, 409.6, 819.2$ shown with different dashed and dotted lines. The inset shows the same data with the time rescaled with $\alpha_C$. For comparison the continuous lines show the correlation with the initial configuration $C(t, 0)$. **(a)** $(2 + s)$-spin models. **(b)** $(3 + s)$-spin models.

### 3.7 What kind of aging do we observe?

As conjectured in Ref. [8] and shown in Ref. [62],[2] the dynamics of the pure 3-spin model presents a simple aging in the asymptotic regime, i.e. two-time observables such as $C(t, t_w)$ depend only on the ratio $(t - t_w)/t_w$. An intermediate sub-aging crossover appears for finite times, only for quenches near the critical temperature $T \lesssim T_{\text{MCT}}$. (The relative scaling can be exactly studied in the case of continuous fullRSB transitions [56]). Because we are considering quenches to $T = 0$, we assume that crossover behavior is suppressed. We observe (Fig. 11) that both $(2 + s)$ and $(3 + s)$ models seem to present simple aging in the asymptotic dynamics, at least over the available time scales.

## 4 Conclusions

In this paper we presented a detailed analysis of the gradient descent dynamics starting from a random initial condition, revisiting and extending previous work [8, 9, 25] to a general class of mixed $(p + s)$-spin glass models.

Our main results are the following (Fig. 12).

- We confirm that, in all cases, the dynamics converges asymptotically to a marginally stable minimum, such that the support of its density of vibrational modes touches zero. Correspondingly, all quantities converge to their asymptotic limits as power laws, with exponents that we estimate from a series expansion [26].

- We confirm that in pure $p$-spin models ($p = s$) the energy converges to the threshold energy $E_{th}$ that separates minima ($E < E_{th}$) from saddles ($E > E_{th}$). (This is a consequence of the fact that in pure models, marginal states only exist at the threshold level [8, 9].) The threshold level is asymptotically uniformly sampled by the dynamics, leading to weak ergodicity breaking, loss of memory of the initial condition, the emergence of a single effective temperature associated to the slow degrees of freedom, and

---

[2] Notice that Ref. [62] presents a numerical integration of the DMFT equations reaching times of order $10^7$. However, their adaptive algorithm is not robust and in particular is unstable if used in the zero temperature GD dynamics, as commented in Ref. [25].

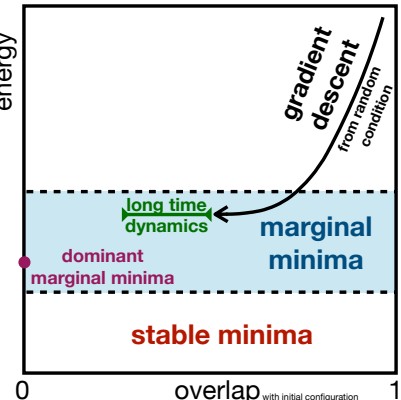

Figure 12: Scheme for the gradient descent dynamics from random initial condition in mixed $(p+s)$-spin spherical models. The system reaches marginal energies above the energy of dominant marginal minima $E_{th}$, and it does not lose memory of the initial condition.

time-reparametrization invariance, as predicted by the asymptotic solution of Cugliandolo and Kurchan [8].

- For mixed $(p+s)$-spin models with $s \gg p$, we obtain quite strong numerical indications against weak ergodicity breaking. The correlation between the initial and final states of the dynamics seem to remain finite, indicating that the dynamics remain confined in a restricted manifold that depends on the initial condition. One should of course keep in mind that our results are limited to finite times, and it is impossible to completely exclude that the dynamics could present a crossover to a weak ergodicity breaking regime at much larger times. We note, however, that such a crossover would already be a quite non-trivial phenomenon that is not captured by current existing theories of the asymptotic aging dynamics. Furthermore, even if weak ergodicity breaking is restored at very large time, the asymptotic manifold cannot be the 1-RSB threshold associated to the Cugliandolo-Kurchan solution, because this energy goes below the ground state energy for large enough $s$.

- Moreover, as far as we can integrate, the extrapolated asymptotic energy lies above the energy at which typical minima become marginal. The convergence to this marginal manifold follows a power-law decay $t^{-\alpha}$, with a power $\alpha \leq 2/3$ that depends on the specific model.

- Yet, our results suggest that the large-time asymptotic dynamics is largely independent of the short-time details, suggesting that despite strong ergodicity breaking, the asymptotic manifold is sampled in some effectively thermal way.

- As in the Cugliandolo-Kurchan solution, the effective temperature function $\chi(q)$ converges to a finite limit for large times, but the asymptotic function does not seem to be described by any known ansatz for the long-time dynamics.

- For models with $s$ close to $p$, such as the $(3+4)$ model studied in Ref. [25], we find that the strong ergodicity breaking, if present, is very weak. It is difficult to decide, but our feeling is that there is strong ergodicity breaking at any $s > p$, which means that the claim made in Ref. [25] of the existence of a finite $T_{\text{onset}}$ separating weak and strong ergodicity breaking might be incorrect, i.e. $T_{\text{onset}} = \infty$. In fact, the semi-phenomenological approximation adopted in Ref. [25] to extract the onset temperature makes use of a 1-RSB

structure of the aging dynamics (one effective temperature), which we have shown to be not valid for large $s - p$.

The coherence of our results suggests that the regime of times we can access is close to the asymptotic one, which calls for a rethinking of the asymptotic dynamics in mean-field models. We believe that our results are compatible with several scenarios:

a) The most likely scenario, in our opinion, is that of strong ergodicity breaking. In that case, one should look for an asymptotic solution with a finite $\tilde{q} = \lim_{t \to \infty} C(t, 0)$. This asymptotic solution would also be characterized, as in the Cugliandolo-Kurchan scheme [24], by a hierarchy of time scales with time-reparametrization invariance [11] and a non-trivial effective function $x[q]$. First steps in this direction have been taken in Refs. [25, 63], but the analysis is far from being complete.

b) The other option is that weak ergodicity breaking ($\tilde{q} = 0$) is restored at large times. As pointed out before, this requires a non-trivial crossover, e.g. to a logarithmic time decay [22, 23]. The corresponding asymptotic solution cannot be that of Cugliandolo and Kurchan with a single slow time scale (1-RSB) [8], because the asymptotic energy corresponding to that solution is the 1-RSB threshold that goes below the ground state for large $s$. Hence, a different solution should be constructed, probably with multiple slow time scales and, again, a non-trivial $x[q]$ [24, 63].

We believe that constructing such a solution is an important problem for future work, because it would shed light on problems such as the mean-field dynamics near to jamming, and the corrections to mean-field aging in finite-dimensional structural glasses [34].

# Acknowledgments

We thank L.Cugliandolo for bringing Ref. [26] to our attention. We also thank A.Altieri, L.Berthier, G.Biroli, P.Charbonneau, L.Cugliandolo, S.Franz, J.Kent-Dobias, J.Kurchan, and F.Ricci-Tersenghi for many useful discussions related to this work. This project has received funding from the European Research Council (ERC) under the European Union's Horizon 2020 research and innovation programme (grant agreement n. 723955 - GlassUniversality).

# A   Numerical integration

In order to systematically study the long time GD dynamics we have numerically integrated Eqs. (16) by a fixed time-step $dt$ algorithm [25, 64]. Each integral in Eqs. (16) is computed with linear order in $dt$, resulting in an integrated dynamics that converges linearly in $dt$ to the exact value ($dt \to 0$). The computation cost of the algorithm scale as $L^3$, $L$ being the size of the 2d-grid of $C(t, t')$ and $R(t, t')$ with discrete time couples $t = i \times dt, t' = j \times dt$. We have used $L = 30000$, which corresponds to 15 GB of RAM and a computing time of 2/3 days on a standard computer. We have chosen $dt = 0.05$ to be a 'good' compromise between $dt \to 0$ convergence and final time $t_f = L dt = 1500$. A consistency check of the results obtained with $dt = 0.05$ is presented in Fig. 13 for the $(3 + 11)$-spin model, where the quadratic extrapolation for $dt \to 0$ given $dt = 0.3, 0.4, 0.5$ is compared with the linear extrapolation given $dt = 0.3, 0.4$ (red line) and with not extrapolated results for each $dt$. We notice that for $dt = 0.05$ the dynamics asymptotically converges to the extrapolated one. For the energy $E$ and the radial reaction $\mu$ this convergence is very fast, instead for the correlation

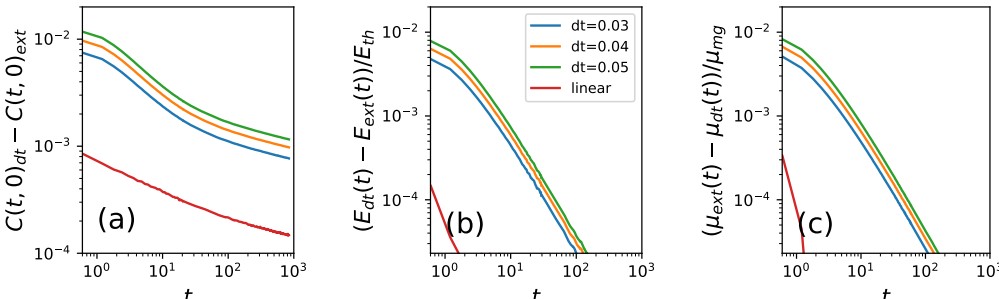

Figure 13: Extrapolation of the integrated dynamics for $dt \to 0$ in the $(3+11)$-spin model. Three different $dt = 0.03, 0.04, 0.05$ are considered. Both a quadratic interpolation and a linear in $dt = 0.03, 0.04$ are evaluated and compared with the non-extrapolated curves. **(a)** The correlation function. **(b)** The energy. **(c)** The radial reaction. The red lines represents the difference between quadratic and linear extrapolation. For each observable, increasing time the error given by the step $dt$ decreases. Notice that for the correlation the decrease is slower. This analysis support the choice of $dt = 0.05$ as a 'good' step for the asymptotic analysis.

$C(t, 0)$ it is slower. However for times $\gtrsim 100$ the error is $\sim 10^{-3}$. Similar behaviour is found for any considered $(2+s)$ and $(3+s)$ model, with a slow increase in the error by increasing $s$.

# B   Series expansion

In this appendix we explain how to evaluate the solution of the DMFT integro-differential equations by finding a recursive relation on the polynomial coefficients of a short-time series expansion, as first suggested in Ref. [26].

## B.1   Equilibrium

We start by the simpler one-dimensional example that corresponds to the equilibrium dynamics,

$$\partial_t C(t) = -C(t) - \beta^2 \int_0^t ds f'(C(s)) \partial_s C(t-s), \qquad (B.1)$$

with $f(q) = \sum_p \alpha_p^2 q^p / 2$. Given a Taylor expansion around $t = 0$, in the form $C(t) = \sum_{k=0}^{\infty} C_k t^k$, this equation reads

$$\sum_{k=0}^{\infty} (k+1) C_{k+1} t^k = -\sum_{k=0}^{\infty} C_k t^k - \beta^2 \int_0^t ds \left( \sum_{m=0}^{\infty} C_m s^m \right)^{p-1} \left( \sum_{l=0}^{\infty} l C_l (t-s)^{l-1} \right), \qquad (B.2)$$

where for simplicity we have considered the pure model of degree $p$. The last term can be rewritten as

$$\int_0^t ds f'(C(s)) \partial_s C(t-s) = \sum_p \alpha_p^2 \sum_k F_k^p t^k,$$

$$F_k^p = \frac{p}{2} \sum_{m_1+m_2+...+m_{p-1}+l=k} C_{m_1} C_{m_2}...C_{m_{p-1}} l C_l \left( \frac{\sum_{q=0}^{l-1} \binom{l-1}{q}(-1)^q}{m_1+m_2+...+m_{p-1}+1} \right) \qquad \text{(B.3)}$$

$$= \frac{p}{2} \sum_m \frac{C_{k-m}}{\binom{k}{m}} \sum_{m_1+m_2+...+m_{p-1}=m} C_{m_1} C_{m_2}...C_{m_{p-1}},$$

where we used the binomial expansion $(t-s)^{l-1} = \sum_{q=0}^{l-1} \binom{l-1}{q} t^{l-1-q}(-s)^q$ and the power expansion

$$\left( \sum_{l=0}^\infty C_m s^m \right)^{p-1} = \sum_{m_1} \sum_{m_2} ... \sum_{m_{p-1}} C_{m_1} C_{m_2}...C_{m_{p-1}} s^{m_1+m_2+...+m_{p-1}}. \qquad \text{(B.4)}$$

The last sum can also be rewritten in an encapsulated form which in a $p=5$ case reads

$$S_m^5 \equiv \sum_{m_{123}=0}^m C_{m-m_{123}} \sum_{m_{12}=0}^{m_{123}} C_{m_{123}-m_{12}} \sum_{m_1=0}^{m_{12}} C_{m_{12}-m_1}, \qquad \text{(B.5)}$$

where $m_{12} = m_1 + m_2$, $m_{123} = m_{12} + m_3$, $m = m_{123} + m_4$. Thus the original equation becomes a recursive equation for the polynomial coefficients

$$C_{k+1} = -\frac{(C_k + \beta^2 \sum_p \alpha_p^2 F_k^p)}{k+1}. \qquad \text{(B.6)}$$

The obtained series has a small radius of convergence in $t$, thus to retain the maximum of information we Padé-approximate it with a rational function of half the degree, as it will be explained in Sec. B.3.

## B.2 Out-of-equilibrium

We now treat the out-of-equilibrium case, in which correlations depend on two times. We first introduce a useful identity. If we need to evaluate the Taylor series of the product $C(t) = A(t)B(t)$, we have

$$C_k = \sum_{i=0}^k A_i B_{k-i}, \qquad \text{(B.7)}$$

where $C(t) = \sum_i C_i t^i$ and similarly for $A(t)$ and $B(t)$. Therefore the power expansion of any power $p$ of a function $C^p(t)$ can be iteratively deduced from the power expansion of its lower degrees, $C_k \longrightarrow C_k^2 \longrightarrow \cdots \longrightarrow C_k^p$, where each iteration has a computational cost $\propto k$. This rule makes it possible to evaluate Taylor series for large values $p$ of the $p$-spin interaction. The same construction can be applied to the product of two-time functions $C(t,t') = A(t,t')B(t,t')$, giving

$$C_{k,l} = \sum_{i=0}^k \sum_{j=0}^l A_{i,j} B_{k-i,l-j}. \qquad \text{(B.8)}$$

In the following we will call total degree $w = k + l$ the sum of the single index degrees. We now proceed to derive the iterative equations that allow one to obtain the two-time Taylor

expansion of $C(t, t')$ and $R(t, t')$ from the out-of-equilibrium Eqs. (16). When we expand in series the integrals, two classes of integrals appear:

$$\int_0^{t'} ds f'(C_{ts}) R_{t's}, \quad \int_0^{t'} ds f''(C_{ts}) R_{ts} C_{t's} \longrightarrow \int_0^{t'} ds A(t,s) B(t',s),$$

$$\int_{t'}^{t} ds f''(C_{ts}) R_{ts} C_{st'}, \quad \int_{t'}^{t} ds f''(C_{ts}) R_{ts} R_{st'} \longrightarrow \int_{t'}^{t} ds A(t,s) B(s,t'), \tag{B.9}$$

where in each two-time function $F(t_1, t_2)$ we always have $t_1 > t_2$. The first class of integrals can be expanded in series as

$$I^{1,AB}(t, t') \equiv \int_0^{t'} ds A(t,s) B(t',s) = \int_0^{t'} ds \sum_{i,j} \sum_{k,l} A_{i,j} t^i s^j B_{k,l} t'^k s^l = \sum_{i,j,k,l} A_{i,j} B_{k,l} t^i t'^k \frac{t'^{j+l+1}}{j+l+1}, \tag{B.10}$$

which gives the coefficients

$$I_{p,q}^{1,AB} = \sum_{i=p} \sum_{k+j+l+1=q} \frac{A_{i,j} B_{k,l}}{j+l+1}. \tag{B.11}$$

The second class of integrals is expanded as

$$I^{2,AB}(t, t') \equiv \int_{t'}^{t} ds A(t,s) B(s,t') = \int_{t'}^{t} ds \sum_{i,j} \sum_{k,l} A_{i,j} t^i s^j B_{k,l} s^k t'^l$$

$$= \sum_{i,j,k,l} A_{i,j} B_{k,l} t^i t'^l \frac{\left(t^{j+k+1} - t'^{j+k+1}\right)}{j+k+1}, \tag{B.12}$$

which gives the coefficients

$$I_{p,q}^{2,AB} = \sum_{i+j+k+1=p} \sum_{l=q} \frac{A_{i,j} B_{k,l}}{j+k+1} - \sum_{i=p} \sum_{l+j+k+1=q} \frac{A_{i,j} B_{k,l}}{j+k+1}. \tag{B.13}$$

Given these two expansions, we are able to express all the integrals in the Eqs. (16) as power series. In the calculation of our series expansion, following Ref. [26], we will proceed by increasing the total degree $w$ in unit steps, as depicted in Fig. 14. We notice that the computational cost of the coefficient of order $w$ of each integral scales as $w^2$.

Finally, for the radial reaction $\mu(t) = T + I_{kl}^{1,f'R}(t,t) + I_{kl}^{1,(f''R)C}(t,t)$, we obtain the series

$$\mu_q = \sum_{k+l=q} I_{kl}^{1,f'R} + \sum_{k+l=q} I_{kl}^{1,(f''R)C}, \qquad \forall q > 0, \tag{B.14}$$

and $\mu_0 = T$.

The final coefficient equations are

$$(k+1) C_{(k+1)l} = - \sum_{k_1+k_2=k} \mu_{k_1} C_{k_2 l} + I_{kl}^{1,f'R} + I_{kl}^{1,(f''R)C} + I_{kl}^{2,(f''R)C}, \tag{B.15}$$

and

$$(k+1) R_{(k+1)l} = - \sum_{k_1+k_2=k} \mu_{k_1} R_{k_2 l} + I_{kl}^{2,(f''R)R}. \tag{B.16}$$

Moreover, we have the constraints $C(t,t) = 1$ and $R(t^+, t) = 1$ for all $t$, which in terms of Taylor coefficients gives

$$\sum_{k+l=w} C_{kl} = 0, \quad \text{and} \quad \sum_{k+l=w} R_{kl} = 0, \quad \forall w > 0. \tag{B.17}$$

These are used to evaluate the terms $C_{0l}$ and $R_{0l}$ not given by Eqs. (B.15) and (B.16). The algorithm runs in increasing order of the degree $w = k + l$, as shown in Fig. 14.

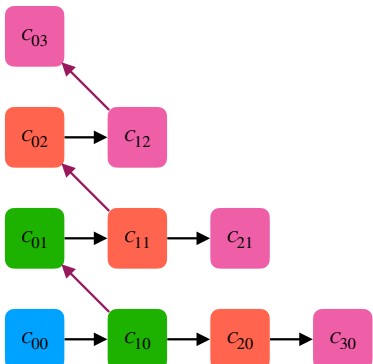

Figure 14: Iteration scheme for the progressive evaluation of the coefficients. Different colors refers to different degrees $w$. A new diagonal is obtain for $C$ and $R$ by Eqs. (B.15) and (B.16) (black arrows) and the constraint in Eq. (B.17) is then implemented (red arrow.)

### B.3 Padé approximation

Once we have the power series in the two times $t'$, $t$, we wish to study its long-time behavior. However, the series have a small radius of convergence (less than one in both times). This usually happens since the true function (the exact solution of the dynamics) has some poles at that distance from the origin in the complex plane.[3] In order to get useful results, as suggested in Ref. [26], we proceed using the Padé approximation, a powerful method to extract the information hidden in the series. It consists in rewriting the original Taylor polynomial of degree $2w$ in terms of a ratio of polynomials of degree $w$ (or of similar degree), in such a way that the Taylor expansion of this ratio is equal to original one. The underlying idea is that the Padé rational function can absorbs the poles of the original function, avoiding the divergence that occurs in the original Taylor series.

We have analysed three main one-time observables:[4] the energy $E(t)$, the radial reaction $\mu(t)$ and the correlation with the initial configuration $C(t,0)$. For every chosen model in the $(2+s)$ and $(3+s)$ class, we have evaluated the first $2w = 1200$ orders and thus a $w = 600$ Padé terms at the numerator and denominator,

$$E(t) \sim \sum_{k=0}^{2w} e_k t^k \sim \frac{\sum_{k=0}^{w} e_k^{\mathrm{n}} t^k}{\sum_{k=0}^{w} e_k^{\mathrm{d}} t^k}, \tag{B.18}$$

where $\sim$ means that they have the same Taylor expansion at $t = 0$. The coefficients $e_k^{\mathrm{n}}, e_k^{\mathrm{d}}$ are evaluated from the $e_k$ by solving a linear equation (one-matrix inversion) [65]. The Padé approximation is very effective in substantially suppressing the influence of the closest poles, and allowing one to reach times $\tau$ (see Fig.15) that grow linearly with the number of terms $2w$ of the original series, roughly as $\tau \approx w/10$. Yet, the values of $\tau$ reached in this way are at least one decade smaller than those reached by numerical integration of the equations with a fixed time step $dt = 0.05$. The main advantage of the series analysis is that it allow us to evaluate the power-law exponents of the decay with much greater precision than the integration, as we

---

[3]It is important to notice that having a Taylor expansion with a given radius of convergence and doing a naive numerical analytical continuation does not give any benefice, in the sense that it is not possible to circumnavigate a pole by finite series expansion, i.e. a closure for the series is needed.

[4]We have only considered the 1-dimensional Padé computation, while in principle it would be possible to extend Padé computations to a multidimensional case, the so-called Canterbury Approximations.

see in the next subsection. In our analysis we have thus employed a hybrid method, using the series to evaluate the exponents and the integration to evaluate the correspondent asymptotic value.

We notice that in order to explore large times, the Padé series needs to have very large powers in $t$, up to $t^{600}$. To obtain meaningful results, the coefficients $e_k$ of the series must thus be evaluated with very high precision. It is not enough to use long-long double precision, as the number of digits considered should scale with the order $w$. Therefore, when numerically evaluating the series (as described in the previous paragraph) we have kept a precision of 2000 digits for each coefficient. This can be achieved by using dedicated multi-precision floating-point libraries. For this work, we have used the c++ library GNU MPFR.

### B.4 Power law evaluation

Despite that the series expansion (even when Padé-transformed) does not allow to reach long times, it can be used to precisely evaluate the power-law decay of different observables to their asymptotic value. Following [26, 66] we define

$$\alpha_E(t) = -\left(1 + \frac{t\partial_t^2 E(t)}{\partial_t E(t)}\right),$$
(B.19)

and by definition $\lim_{t\to\infty} \alpha_E(t) = \alpha_E$ defined in Eq. (18). Analogous definitions can be given for $\alpha_\mu(t)$ and $\alpha_C(t)$. Given the Taylor series for $E(t)$, we evaluate $\alpha_E(t)$ and Padé-transform it. The results for the energy of $(3+s)$-models are shown in Fig. 15. Note that for a precise analysis of the exponents, it is not convenient to analyze directly the Padé transform of the original series, since by construction the Padé approximation is a rational function that can only asymptotically behave as an integer power law, while we want to study non-integer power-law decays. Finally, we fit each $\alpha(t)$ with an exponential function (dashed lines in Fig. 15) and extract the asymptotic value that correspond to the power-law exponent introduced in Eq. (18). In table 2 we report the values of the exponents in the $(2+s)$ and $(3+s)$ models for the values of $\lambda$ given in table 1. We can check the accuracy of the extrapolation in the pure 2-spin model for which we have exact analytical results [40]. The energy exponent is known to be $\alpha_E = \alpha_\mu = 1$, and we obtain $0.999 \pm 0.001$, while the correlation is known to have exponent $\alpha_C = 3/4$ and we obtain $0.748 \pm 0.001$. For the pure 3-spin model the exponents seem to agree with the values $\alpha_E = \alpha_\mu = 2/3$ and $\alpha_C = 3/8$. Mixed models have variable exponents smaller than the pure ones, as could be expected because the interplay between different interactions slows down the dynamics. Since our analysis of the exponents is carried on for not vary large times (smaller than 100), we cannot exclude that other regimes could emerge for later times. Yet, the monotonous behaviour of the $\alpha(t)$ functions for $t > 10$ for any $s$ seems to support the non-universality of exponents in the relaxation of mixed $p$-spin models.

## C  Thermodynamic solution for the dynamics

In order to understand the dynamics in terms of a static (thermodynamic) calculation we start from the free energy of the mixed $p$-spin model calculated on a generic Parisi hierarchical ansatz for the replica overlap matrix $Q_{ab}$. This is conveniently written in functional form in terms of the susceptibility $\tilde{\chi}(q) = \chi(q)/\beta$, with the additional boundary conditions $\tilde{\chi}(1) = 0$

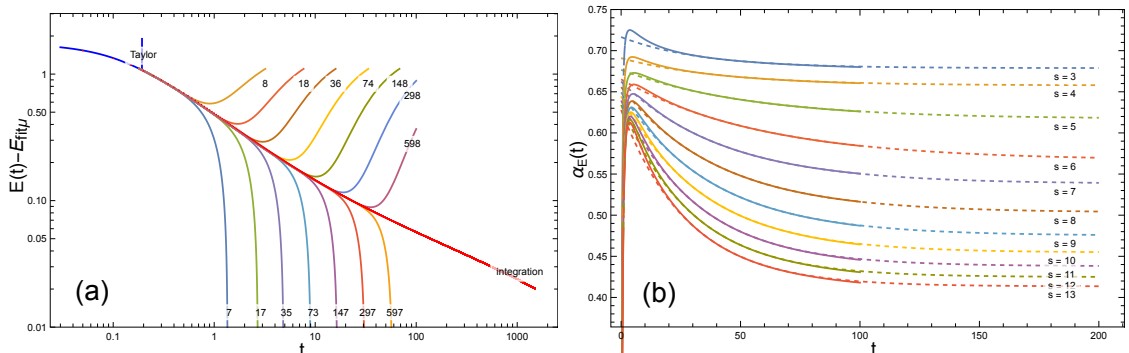

Figure 15: **(a)** Padé approximation of the energy $E(t)$ in the (3+11)-spin model. The inferred asymptotic energy $E_{\text{fit}\mu}$ is subtracted to show the power-law decay. Different orders $w$ (from 7 to 598, $\approx 1200/2^k$ with $k$ from 1 to 7) of Padé approximations are shown in different colors. These are compared with the solution obtained by integrating the equations with a finite step size $dt = 0.05$ (red points). The Taylor series has a radius of convergence $< 0.3$. We observe that the order of the Padé approximation is directly proportional to the time $\tau$ below which convergence is observed (equi-spaced colored lines in log scale). Because the computational complexity of the series and its Padé approximation scales as $w^3$, the computational complexity of the integration and of the series both scales as $\tau^3$. **(b)** Time dependence of $\alpha_E(t)$ in (3+$s$)-spin models, for different $s$. The dashed lines represents exponential fits in the range $(30, 50)$. From this fit the asymptotic value correspondent to the power-law decay is extracted.

and $\tilde{\chi}(1)' = -1$ [39,49],

$$
\begin{aligned}
F[\chi] = E[\chi] - TS[\chi] &= \frac{1}{2}\int_0^1 dq \left( f''(q)\beta\tilde{\chi}(q) + (\beta\tilde{\chi}(q))^{-1} \right) \\
&= \frac{1}{2}\int_0^1 dq\, f''(q)^{1/2}\left( \hat{\chi}(q) + \hat{\chi}(q)^{-1} \right), \\
E[\chi] = \frac{\partial \beta F}{\partial \beta} &= \int_0^1 dq\, f''(q)\beta\tilde{\chi}(q) = \int_0^1 dq\, f''(q)^{1/2}\hat{\chi}(q), \\
S[\chi] = -\frac{\partial F}{\partial T} &= \frac{1}{2}\int_0^1 dq \left( f''(q)\beta^2\tilde{\chi}(q) - (\tilde{\chi}(q))^{-1} \right).
\end{aligned}
\tag{C.1}
$$

Because the boundary conditions on $\tilde{\chi}(q)$ do not depend on $\beta$, the thermal derivatives can be taken only over the explicit temperature dependence, while the implicit dependence on $\tilde{\chi}$ does not contribute because $\partial F / \partial \tilde{\chi} = 0$. Then, we can change variable to $\chi(q) = \beta\tilde{\chi}(q)$ with boundary conditions $\chi(1) = 0$ and $\chi(1)' = -\beta$, which is an implicit statement of FDT at equilibrium. We have also introduced another scaled function $\hat{\chi}(q) = \chi(q)f''(q)^{1/2}$, which is sometimes more convenient. The function $\chi(q)$ is in a one-to-one bijection with the Parisi matrix $Q_{ab}$ (corresponding to its eigenvalues). For example considering a piecewise linear function

$$
\chi(q) = \chi_0, \, q \in [0, q_0]; \quad \chi_1 + m(q_1 - q), \, q \in [q_0, q_1]; \quad (1 - q), \, q \in [q_1, 1], \tag{C.2}
$$

gives back the 1-RSB free energy, Eq. (25) of Ref. [39]. The first term in Eq. (C.1) is the energy $E[\chi] = \int_0^1 dq f''(q)\chi(q)$ and it is equal to the dynamic definition in Eq. (17) if we consider

$\chi(q)$ as the integrated response. The second term $S[\chi]$ corresponds instead to the entropic contribution, roughly given by the logarithm of the volume of a sphere of radius corresponding to the overlap $q$. This second term does not have a clear correspondent in the dynamics, and we believe it to be responsible for the discrepancy between the static calculation and the asymptotic limit of the dynamics. Moreover $x(q) = -\partial_q \chi(q)$ is the fluctuation-dissipation ratio and $P(q) \propto \partial_q x(q) = -\partial_q^2 \chi(q)$ is the probability of finding two states at overlap $q$. We see that in order to have a positive probability, the second derivative of $\chi(q)$ must be negative (concave) and its first derivative negative (monotonic). Thus in the functional space of all regular $\chi(q)$ only concave-monotonic ones must be considered when minimizing Eq. (C.1).

In the zero temperature limit ($\beta \to \infty$) the boundary conditions are trivially satisfied by a jump from $\chi(1) = 0$ to a finite value $\chi(1^-)$. This jump does not contribute to Eq. (C.1) and can thus be discarded. The ground state corresponds to the optimum between all concave solutions. There exists however other possible solutions, corresponding to non-optimal states, that can be metastable, i.e. have a higher free energy while being locally stable.

An important non-thermodynamic solution is obtained by optimizing Eq. (C.1) without any constraint on the convexity of $\chi(q)$,

$$2\frac{\delta F[\chi(q)]}{\delta \chi(q)} = \int_0^1 dq \left( f''(q) - \chi(q)^{-2} \right) = \int_0^1 dq \, f''(q) \left( 1 - \hat{\chi}(q)^{-2} \right) = 0, \qquad \text{(C.3)}$$

which is identically satisfied by the so-called (non-strictly-concave) algorithmic solution $\chi_{alg}(q) = f''(q)^{-1/2}$, or equivalently $\hat{\chi}_{alg}(q) = 1$. Note that $\partial_q \chi_{alg}(q) < 0$ for $q \in [0,1]$, because $f(q)$ belongs to the class of polynomials with positive coefficients. Its corresponding energy is exactly the algorithmic energy $E_{alg}$ defined in Eq. (13). All the other solutions (concave or not) will have a higher free energy, but they can have a smaller energy, as it is the case for the ground state solution.

We now wish to find a concave solution that corresponds to what is observed in the GD dynamics. The first observation is that the GD dynamics is asymptotically marginal, i.e. $\chi(1^-) = \chi_{mg} = f''(1)^{-1/2}$. This is our additional constraint in the search for a concave solution. The classical solution is the so-called 1-RSB dynamical solution derived by Cugliandolo and Kurchan [8]. This solution can be found in the static concave-minimization scheme by following the Monasson construction [67], which consists in (i) postulating a linear solution (thus quasi-concave and monotonic) $\chi^{(1)}(q) = \chi + x(1-q)$, (ii) inserting it in Eq. (C.1), which gives

$$2F[\chi^{(1)}] = f'(1)\chi + xf(1) + \frac{1}{x} \log\left( \frac{\chi + x}{\chi} \right), \qquad \text{(C.4)}$$

(iii) extremizing it with respect to $\chi$ at fixed $x$ (the effective temperature) obtaining $x^*(\chi) = \frac{1}{\chi f'(1)} - \chi$ and (iv) imposing the marginal condition $\chi = \chi_{mg}$, thus obtaining the energy of the marginal state $E[\chi^{(1)}] = f'(1)\chi_{mg} + x^*(\chi_{mg})f(1)$, which is indeed the same as Eq. (12). In this solution, $x^*(\chi) = \partial\Sigma(\chi)/\partial E(\chi)$ is the temperature associated to the complexity of metastable states of given linear susceptibility $\chi$. Notice that minimizing Eq. (C.4) with respect of both $\chi$ and $x$ gives the 1-RSB ground-state solution.

In the case of a pure $p$-spin model with $p > 2$, this is the only possible solution, since the correspondent $\chi_{alg}(q)$ is non-concave everywhere. Hence (as argued in [41]) there cannot be any (meta)stable solution with more then 1-RSB. The only possibility of having more complicated solutions comes from the presence of concave sectors in the algorithmic solution $\chi_{alg}(q)$, i.e. if it exists some $q \in [0,1]$ such that $\partial_q^2 \chi_{alg}(q) < 0$, or equivalently

$$3qf'''(q)^2 - 2f''''(q)f''(q) < 0. \qquad \text{(C.5)}$$

In the case of the $(2+s)$ models there always exists such a concave sector near $q = 0$. Instead in the selected $(3+s)$ models a concave sector develops in the middle of the $q$-interval $[0,1]$ for

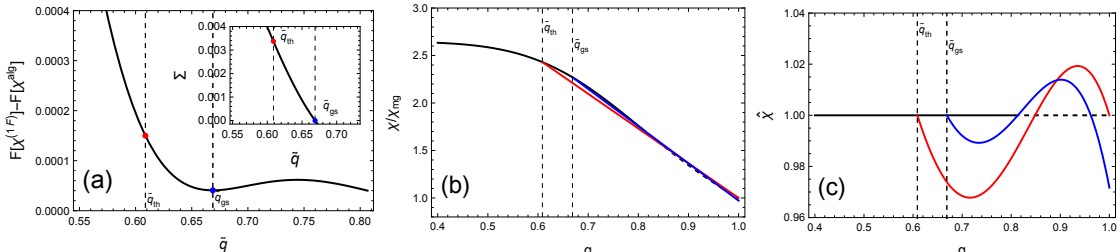

Figure 16: Marginal 1F solution in the (2+9) model. **(a)** Free energy $F[\chi^{(1F)}]$ of the 1F solution vs the point of contact $\tilde{q}$. The ground state corresponds to the global minimum $\tilde{q}_{gs}$. The inset shows the relative complexity $\Sigma = (1/x^2)\partial_x F[\chi^{(1F)}]$. **(b)** $\chi/\chi_{mg}$ as a function of $q$ for the marginal $\tilde{q}_{th}$ (red line) and the ground-state $\tilde{q}_{gs}$ (blue line). The black line shows the algorithmic solution $\chi^{(1F)}$ with the non-concave sector shown as a dashed line (more visible in panel c). **(c)** $\hat{\chi} = \chi/\chi^{alg}$ as a function of $q$. Same curves as in panel b.

large enough $s$ only. Whenever there is a concave sector, it may be possible –also considering the monotonicity constraint– to build alternative solutions to the 1-RSB marginal one.

Let us now consider another possible marginal ansatz, namely the 1F "dynamical" solution [39], which is a possible solution in $(2+s)$ models. This is defined by a collage of a 1-RSB and a fullRSB solution, $\chi^{(1F)}(q) = \chi^{alg}_{[0,\tilde{q}]}(q) + \chi^{(1)}_{[\tilde{q},1]}(q)$, where the sub-parenthesis indicates the interval of $q$ in which each solution is considered. Inserting this ansatz in Eq. (C.1), we obtain

$$2F[\chi^{(1F)}] = \int_0^{\tilde{q}} dq\, f''(q)^{1/2} - \tilde{\chi}f'(\tilde{q}) + \chi f'(1) + x(f(1) - f(\tilde{q})) + \int_0^{\tilde{q}} dq\, f''(q)^{1/2} + \frac{1}{x}\log\left(\frac{\tilde{\chi}}{\chi}\right),$$

(C.6)

where $\tilde{\chi} = f''(\tilde{q})^{-1/2}$ and $\chi = f''(\tilde{q})^{-1/2} + x(\tilde{q} - 1)$, hence the linear part is given by $\chi^{(1)}_{[\tilde{q},1]}(q) = \tilde{\chi} - x(q - \tilde{q})$. Therefore $F[\chi^{(1F)}]$ is a function of the two parameters $\tilde{q}$ and $x$. Minimizing over both of them gives the 1F ground-state. Instead, in order to build the metastable "dynamical" 1F solution we follow the Monasson construction described above. We minimize with respect to $\tilde{q}$ while keeping fixed the effective temperature $x$, thus obtaining[5]

$$x^*(\tilde{q}) = \frac{\frac{1}{1-\tilde{q}} - \frac{f''(\tilde{q})}{f'(1) - f'(\tilde{q})}}{\sqrt{f''(\tilde{q})}}.$$

(C.7)

Substituting it back in Eq. (C.6) we obtain the $F[\chi^{(1F)}]$ as a function of $\tilde{q}$, as shown in Fig. 16a. Finally, imposing the marginality condition $\chi = \chi_{mg} = f''(1)^{-1/2}$ gives the equation $f''(\tilde{q})^{-1/2} + x^*(\tilde{q})(\tilde{q} - 1) = f''(1)^{-1/2}$ which fixes the threshold overlap

$$\tilde{q}_{th} \quad \text{s.t.} \quad \frac{(\tilde{q} - 1)\sqrt{f''(\tilde{q})}}{f'(\tilde{q}) - f'(1)} = \frac{1}{\sqrt{f''(1)}},$$

(C.8)

where the use of the term threshold is made in analogy with the Cugliandolo-Kurchan picture. In fact at $\tilde{q}_{th}$ we have the maximal complexity for typical minima. The correspondent threshold energy is

$$E_{th}^{(1F)} \equiv E[\chi^{(1F)}] = \int_0^{\tilde{q}_{th}} dq\, f''(q)^{1/2} - f''(\tilde{q}_{th})^{-1/2}f'(\tilde{q}_{th}) + f''(1)^{-1/2}f'(1) + x^*(\tilde{q}_{th})(f(1) - f(\tilde{q}_{th})).$$

(C.9)

---

[5]A second solution $x^*(\tilde{q}) = \frac{f^{(3)}(\tilde{q})}{2f''(\tilde{q})^{3/2}} \equiv \chi'_{alg}(\tilde{q})$ —which correspond to $\chi^{(1)}$ being tangent to the algorithmic solution $\chi_{alg}$— appears, but it is locally unstable.

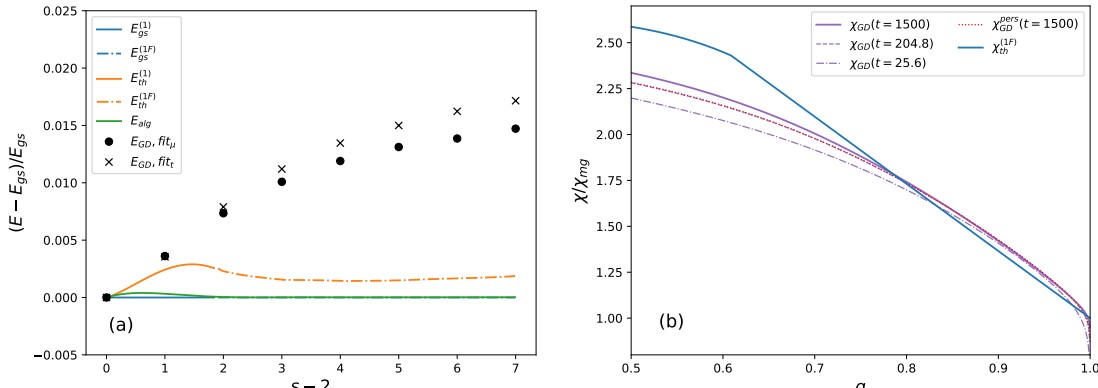

Figure 17: **(a)** Asymptotic energy reached by the gradient descent (GD) dynamics from random initial condition, compared to the ground state energy $E_{gs}$, the threshold energy $E_{th}^{(1)}, E_{th}^{(1F)}$ and the algorithmic energy $E_{alg}$. Same plot as Fig. 2a, but with the value of the threshold energy evaluated with the 1F marginal solution for $s > 3$. **(b)** FDR for the $(2+9)$ model as in Fig. 10a but compared with a 1F marginal solution, i.e. parametric plot over waiting time $t_w$ of the re-scaled integrated response $\chi(t, t_w)/\chi_{mg}$ vs the correlation $C(t, t_w)$, for several fixed values of time $t = 25.6, 204.8, 1500$, compared with the dynamical ansatz ($\chi_{th}^{(1F)}$).

In Fig.16a, the free energy of the dynamical solution $F[\chi^{(1F)}]$ is plotted as a function of $\tilde{q}$ for the $(2+9)$ model. It has a local minimum at the ground state solution $\tilde{q}_{gs}$ (blue point) which correspond to a vanishing complexity (see inset). Instead at $\tilde{q}_{th}$ the solution has maximal complexity (higher-energy solutions are unstable). The corresponding shape of $\chi(q)$ for both $\tilde{q}_{gs}$ (blue) and $\tilde{q}_{th}$ (red) is shown in Fig. 16b and with the re-scaled $\hat{\chi}(q)$ in Fig. 16c. By looking at Fig. 16c we note that (meta)stable solutions must lie both above and below $\chi^{alg}$, and the regions must "compensate", as in the usual Maxwell construction for first order transitions. In Fig. 17a the 1F energy is compared with the asymptotic GD energy for all $(2+s)$ models, while in Fig. 17b we compare the shape of the 1F solution with the GD results. It is evident that this thermodynamic solution does not agree with the GD one.

If we now consider $(3+s)$ models, we find that $\chi_{alg}(q)$ has two non-concave sectors, one near $q = 0$ and the other near $q = 1$, which must be replaced by linear regions to satisfy the convexity requirement. We would then like to consider a solution of the kind $1F1$ as an alternative to the standard 1-RSB solution, i.e. a collage of linear+full+linear. We found that such a solution is not locally stable (see Fig. 18); the full part vanishes and we are left with either a standard 1-RSB solution or at most a 2-RSB one. We will not explore here all the steps of the calculation, but additional comments about the concave minimization of Eq. (C.1) can be found in section 2.1.7 of Ref. [49].

We conclude that minimizing the free energy in Eq. (C.1) in the space of monotonic and concave $\chi(q)$ with the additional constraint of marginality is not consistent with the solution we found from the GD dynamics (see Fig. 17), in particular because the $\chi_{GD}(q)$ obtained from GD dynamics has a shape near $q = 1$ that is not linear, but concave, see Fig. 10 and Fig. 17b. Such a concave shape is not achievable by minimizing the functional in Eq. (C.1). If we still want to find a static (or geometric) description of the asymptotic non-equilibrium dynamics, one possibility would be to modify the entropic part of Eq. (C.1). How to do that remains, however, an open problem. A possible suggestion could come from exploring the quasi-equilibrium dynamics [58, 61].



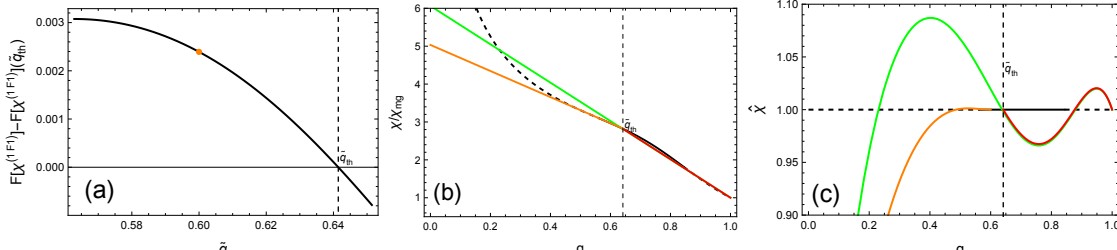

Figure 18: Attempt to build a marginal 1F1 solution in the (3+12) model. **(a)** Free energy $F[\chi^{(1F1)}]$ of the 1F1 solution vs the point of contact $\tilde{q}$ of the left linear branch. $\tilde{q}_{th}$ indicates the point of contact of the right linear branch. Any point of contact of the left linear branch turns out to be unstable, thus the fullRSB continuous region shrinks and eventually vanishes. The orange point indicate the solution plotted in panels b and c. **(b)** $\chi/\chi_{mg}$ as a function of $q$ for a 1F1 unstable solution. The right branch (red line) is stable while the left branch (orange line) is unstable. In green the stable marginal 1RSB solution. **(c)** $\hat{\chi} = \chi/\chi^{alg}$ as a function of $q$. Same curves as in panel b.

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
