# Peer review of "On weak ergodicity breaking in mean-field spin glasses"

_SciPost Physics, doi:SciPost Phys. 15, 109 (2023)_

## Round 1 · Referee Report · Anonymous (Referee 1) · 2023-4-26

Report

The authors analyze mixed spherical random (p+s)-spin glass models (p=2 and 3) undergoing gradient descent dynamics from random initial condition. Numerical integration of the DMFT equations suggests that the weak ergodicity breaking hypothesis does not hold in the mixture models under consideration, at variance with the pure p-spin case. This observation is confirmed by a time series expansion of the overlap with the initial condition C(t,0), reaching a non-zero asymptotic value. A similar expansion of the radial reaction shows that the dynamics approaches a marginally stable minimum, while the asymptotic energy remains higher than the threshold value at which typical minima become marginal. Overall, these results suggest that strong ergodicity breaking is verified in mixed (p+s)-spin models at any s>p and even from infinite initial temperature, in contradiction with the result of ref. [25] where strong ergodicity breaking was found only below a finite initial onset temperature. The dynamics appears to find an aging state confined to an initialization-dependent manifold sampled in an effectively thermal way, as shown by the presence of an effective FDR.

This work inscribes in the timely effort to understand and elucidate a long-standing picture of the out-of-equilibrium dynamics of prototypical mean-field models of the glass transition. The previous literature appears to be cited correctly. The paper is of good technical quality and the analytical derivations are well explained and relatively easy to follow. The presentation is generally good, although some aspects could be improved or clarified (see comments below). For these reasons, I recommend publication of the manuscript in SciPost provided the comments below are taken into account.

Comments and questions:

  • The analytic derivations are written having an expert reader in mind and frequently referring to other works, sometimes to the detriment of a self-contained presentation. E.g., the “overlap” and the “characteristic polynomial” are introduced on page 3 without definition, and similarly the “complexity” on page 4.

  • The coefficients of the mixture seem to play an important role in discriminating between different asymptotic regimes. It would be useful to clarify how sensitive the discrepancy observed, e.g., in Fig. 2 is to the choice of the coefficient \lambda. It would be useful to report the values of the coefficients used in ref. [25] for comparison.

  • It would be interesting if the authors could comment on the implications of their findings for optimization algorithms in planted models, and in relation to the expected discrepancy in performance (if any) between random and “smart” initializations. E.g., should the picture presented in [a,b] be revisited (in particular regarding the (2+4) planted model)? See references below.

  • Ref. [25] supports the finding of a finite onset temperature referring to numerical experiments of realistic glass-forming liquids, where this threshold is observed. How do the authors reconcile this experimental observation with their findings? Does the experimental T_onset show any finite-size dependence? Moreover, according to the phase diagram in Fig. 1(b) of this paper, the 1RSB dynamical ansatz used in ref. [25] to derive T_onset=0.91 for the (3+4)-spin model does not seem unreasonable, in addition to the good agreement with numerical extrapolation. A clarification on why the asymptotic approximation used in [25] is wrong and T_onset instead should be set to infinity would be really appreciated. 

  • The DMFT equations (14) starting from random initialization do not exhibit any explicit dependence on the initial condition via C(t,0). From the presentation in section C, it is not intuitive to me why this dependence should appear in the asymptotic dynamics and I would appreciate a clarification on this point. Moreover, at first one may wonder why the CK ansatz does not apply in this case. I think this point is further clarified in section F, however it could be hard for non-expert readers to connect the dots between sections.

[a] Mannelli, S. S., Biroli, G., Cammarota, C., Krzakala, F., Urbani, P., & Zdeborová, L. (2020). Marvels and pitfalls of the langevin algorithm in noisy high-dimensional inference. Physical Review X, 10(1), 011057.

[b] Sarao Mannelli, S., Biroli, G., Cammarota, C., Krzakala, F., & Zdeborová, L. (2019). Who is afraid of big bad minima? analysis of gradient-flow in spiked matrix-tensor models. Advances in Neural Information Processing Systems, 32.

  • validity: top
  • significance: high
  • originality: high
  • clarity: high
  • formatting: excellent
  • grammar: excellent

Author:  Giampaolo Folena  on 2023-05-09  [id 3657]

(in reply to Report 1 on 2023-04-26)

** Referee**

The authors analyze mixed spherical random (p+s)-spin glass models (p=2 and 3) undergoing gradient descent dynamics from random initial condition. Numerical integration of the DMFT equations suggests that the weak ergodicity breaking hypothesis does not hold in the mixture models under consideration, at variance with the pure p-spin case. This observation is confirmed by a time series expansion of the overlap with the initial condition C(t,0), reaching a non-zero asymptotic value. A similar expansion of the radial reaction shows that the dynamics approaches a marginally stable minimum, while the asymptotic energy remains higher than the threshold value at which typical minima become marginal. Overall, these results suggest that strong ergodicity breaking is verified in mixed (p+s)-spin models at any s>p and even from infinite initial temperature, in contradiction with the result of ref. [25] where strong ergodicity breaking was found only below a finite initial onset temperature. The dynamics appears to find an aging state confined to an initialization-dependent manifold sampled in an effectively thermal way, as shown by the presence of an effective FDR.

This work inscribes in the timely effort to understand and elucidate a long-standing picture of the out-of-equilibrium dynamics of prototypical mean-field models of the glass transition. The previous literature appears to be cited correctly. The paper is of good technical quality and the analytical derivations are well explained and relatively easy to follow. The presentation is generally good, although some aspects could be improved or clarified (see comments below). For these reasons, I recommend publication of the manuscript in SciPost provided the comments below are taken into account.

Response We thank the reviewer for the positive feedback. We have carefully taken into account all provided comments as reported below.

** Referee**

Comments and questions:

  • The analytic derivations are written having an expert reader in mind and frequently referring to other works, sometimes to the detriment of a self-contained presentation. E.g., the “overlap” and the “characteristic polynomial” are introduced on page 3 without definition, and similarly the “complexity” on page 4.

Response We have modified the text to properly introduce the “overlap”, the “characteristic polynomial” and the “complexity”.

** Referee**

  • The coefficients of the mixture seem to play an important role in discriminating between different asymptotic regimes. It would be useful to clarify how sensitive the discrepancy observed, e.g., in Fig. 2 is to the choice of the coefficient \lambda. It would be useful to report the values of the coefficients used in ref. [25] for comparison.

Response We added a footnote to reiterate on the fact that $\lambda$ is chosen in such a way to maximize the discrepancy, and thus the results of ref. [25] shows an even smaller discrepacy.

** Referee**

  • It would be interesting if the authors could comment on the implications of their findings for optimization algorithms in planted models, and in relation to the expected discrepancy in performance (if any) between random and “smart” initializations. E.g., should the picture presented in [a,b] be revisited (in particular regarding the (2+4) planted model)? See references below.

Response Unfortunately, at present we do not have any clear implication of our results on the suggested optmization questions. However, we thank the referee for the suggestion and we will think about this interesting question.

** Referee**

  • Ref. [25] supports the finding of a finite onset temperature referring to numerical experiments of realistic glass-forming liquids, where this threshold is observed. How do the authors reconcile this experimental observation with their findings? Does the experimental T_onset show any finite-size dependence? Moreover, according to the phase diagram in Fig. 1(b) of this paper, the 1RSB dynamical ansatz used in ref. [25] to derive T_onset=0.91 for the (3+4)-spin model does not seem unreasonable, in addition to the good agreement with numerical extrapolation. A clarification on why the asymptotic approximation used in [25] is wrong and T_onset instead should be set to infinity would be really appreciated.

Response T_onset is experimentally a crossover temperature difficult to sharply characterize. In Ref.[25] a sharp T_onset is found because the introduced semi-phenomenological closure of DMFT equations predicts a sharp transition. However, it is a 'good' approximation for the asymptotic dynamics and not an exact result. This approximation is based on the relative closeness of (3+4)-model with pure models. However when the two terms of the mixture (p+s) become more different this approximation gets worse. We have added a phrase to clarify this issue. We feel that numerically, distinguishing between a sharp transition at a finite T_onset or a smoother crossover would be impossible.

** Referee**

  • The DMFT equations (14) starting from random initialization do not exhibit any explicit dependence on the initial condition via C(t,0). From the presentation in section C, it is not intuitive to me why this dependence should appear in the asymptotic dynamics and I would appreciate a clarification on this point. Moreover, at first one may wonder why the CK ansatz does not apply in this case. I think this point is further clarified in section F, however it could be hard for non-expert readers to connect the dots between sections.

Response We do not have any intuition about why the asymptotic dynamics keeps memory of the initial condition without the direct presence of any explicit correlation term C(t,0). It is one of the major questions raised by this paper, and we hope to have risen the attention of the interested audience on this open problem.

[a] Mannelli, S. S., Biroli, G., Cammarota, C., Krzakala, F., Urbani, P., & Zdeborová, L. (2020). Marvels and pitfalls of the langevin algorithm in noisy high-dimensional inference. Physical Review X, 10(1), 011057.

[b] Sarao Mannelli, S., Biroli, G., Cammarota, C., Krzakala, F., & Zdeborová, L. (2019). Who is afraid of big bad minima? analysis of gradient-flow in spiked matrix-tensor models. Advances in Neural Information Processing Systems, 32.

---

## Round 1 · Referee Report · Anonymous (Referee 3) · 2023-4-27

Report

The manuscript address the validity of scenario that has become part of the Spin-Glass folklore in the last thirty years: the idea that off-equilibrium dynamics in mean-field spin-glasses with one step of Parisi's Replica-Symmetry-Breaking explores in a thermal fashion the most numerous marginal states. This idea came from observations on pure p-spin models by Cugliandolo and Kurchan (CK) and is now challenged by recent work including notably this manuscript.

More precisely the authors address the topic of weak versus strong ergodicity breaking, namely if the correlation with initial condition is lost during aging. Based on their results they favor a strong ergodicity breaking scenario while correctly acknowledging that the time regime explored is limited and thus the question cannot be considered settled: "The possibility that the scenario we propose is only a pre-asymptotic regime that would crossover to a weak ergodicity regime thus remains open."

The manuscript discusses a number of features and interesting questions
in a very clear fashion and it is carefully written. It deserves publication
in its present form and I only have a few recommendation.

Quite simply the problem is wether the correlation computed at finite time extrapolate to zero or not at infinite time. If one uses power-laws to extrapolate a finite value is obtained but there is no guarantee that the asymptotic behavior is described by a power-law and not by slower logarithmic decays. Two essential open problems are i) the lack of an analytic solution and ii) the lack of a numerical algorithm capable of reaching large times using a time grid with varying spacing.
Considering a different type of microscopic dynamics, the author show that large times dynamics seems to be independent of the short time details and
this indeed gives hope that an analytic solution can indeed be found and that some of the properties of the CK solution remains valid.

As for the second issue, the authors cite Ref. 60, where one such algorithm was used to reach times of order 10^7, and the reader may be puzzled by why they used a fixed spacing algorithm reaching times of order 10^3, a comment on this seems appropriate.

Ref. 22 is misquoted in the introduction as supporting strong ergodicity breaking but it actually takes an agnostic point of view pointing to the difficulties of extrapolating to large times and urging for an analytical solution. It would be interesting to plot the data of fig. 5 and fig. 6 parametrically and see what would be the asymptotic energy if weak ergodicity breaking was correct, as done in fig. 7 of Ref. 22.

After equation (16) and again in Section G it is mentioned that:" In the case of a quench to the critical temperature, exact relations between the αE and αC exponents were found in Ref. [54]." It is true that Ref. 54 is at present the only case beyond the p=2 spherical model where the asymptotic behavior of one-time quantities is computed analytically, nonetheless it should be stated that it deals with systems with continuous RSB transitions and not with the discontinuous 1RSB systems studied here.

Also, the comment of Dr. Theo Nieuwenhuizen should be taken into account.
  • validity: -
  • significance: -
  • originality: -
  • clarity: -
  • formatting: -
  • grammar: -

Author:  Giampaolo Folena  on 2023-05-09  [id 3659]

(in reply to Report 3 on 2023-04-27)

** Referee**

The manuscript address the validity of scenario that has become part of the Spin-Glass folklore in the last thirty years: the idea that off-equilibrium dynamics in mean-field spin-glasses with one step of Parisi's Replica-Symmetry-Breaking explores in a thermal fashion the most numerous marginal states. This idea came from observations on pure p-spin models by Cugliandolo and Kurchan (CK) and is now challenged by recent work including notably this manuscript.

More precisely the authors address the topic of weak versus strong ergodicity breaking, namely if the correlation with initial condition is lost during aging. Based on their results they favor a strong ergodicity breaking scenario while correctly acknowledging that the time regime explored is limited and thus the question cannot be considered settled: "The possibility that the scenario we propose is only a pre-asymptotic regime that would crossover to a weak ergodicity regime thus remains open."

The manuscript discusses a number of features and interesting questions in a very clear fashion and it is carefully written. It deserves publication in its present form and I only have a few recommendation.

Response We thank the reviewer for the positive report. We have modified the paper according to the provided recommendations.

** Referee**

Quite simply the problem is wether the correlation computed at finite time extrapolate to zero or not at infinite time. If one uses power-laws to extrapolate a finite value is obtained but there is no guarantee that the asymptotic behavior is described by a power-law and not by slower logarithmic decays. Two essential open problems are i) the lack of an analytic solution and ii) the lack of a numerical algorithm capable of reaching large times using a time grid with varying spacing. Considering a different type of microscopic dynamics, the author show that large times dynamics seems to be independent of the short time details and this indeed gives hope that an analytic solution can indeed be found and that some of the properties of the CK solution remains valid.

As for the second issue, the authors cite Ref. 60, where one such algorithm was used to reach times of order 10^7, and the reader may be puzzled by why they used a fixed spacing algorithm reaching times of order 10^3, a comment on this seems appropriate.

Response We added a footnote to comment about this point, as we believe that the algorithm of Ref.60 is not reliable at long times.

** Referee**

Ref. 22 is misquoted in the introduction as supporting strong ergodicity breaking but it actually takes an agnostic point of view pointing to the difficulties of extrapolating to large times and urging for an analytical solution. It would be interesting to plot the data of fig. 5 and fig. 6 parametrically and see what would be the asymptotic energy if weak ergodicity breaking was correct, as done in fig. 7 of Ref. 22.

Response We thank the referee for pointing out this misquotation. We rephrased the sentence.

We have tried the suggested procedure, i.e. plotting the excess energy versus C(t,0) and performing a 3-parameters fit (see attached figure CorrEn.pdf: (a) for 2-spin and (b) for 3-spin). However, the results remain more consistent with the strong ergodicity breaking scenario, without adding further insights. We thus prefer to not add them to the paper, in order to avoid overcharging it.

** Referee**

After equation (16) and again in Section G it is mentioned that:" In the case of a quench to the critical temperature, exact relations between the αE and αC exponents were found in Ref. [54]." It is true that Ref. 54 is at present the only case beyond the p=2 spherical model where the asymptotic behavior of one-time quantities is computed analytically, nonetheless it should be stated that it deals with systems with continuous RSB transitions and not with the discontinuous 1RSB systems studied here.

Response We have rephrased the two phrases that reference [54] in order to clarify that the considered transition is of continous type.

** Referee**

Also, the comment of Dr. Theo Nieuwenhuizen should be taken into account.

Response We have added the suggested reference.

Attachment:

CorrEn.pdf

---

## Round 2 · Referee Report · Anonymous (Referee 3) · 2023-5-24

Report

As I said in the previous report the paper deserves publication. However I ask the authors to make a small additional effort to address an important technical issue. The key ingredient of the paper is the numerical solution of the off-equilibrium dynamical equations by time discretization. Therefore the outcome depends on the time spacing dt. The correct procedure is then to extrapolate the solution to dt=0 considering different values of dt's. It seems that instead the authors report only data at fixed dt=.05 and do not mention the issue at all.
Take for instance fig. 13 (a), it suggests that the numerical curve with dt=.05 does not have large corrections at least up to time t=100 due to the agreement with the results from the series expansion, BUT this is not enough to argue that the agreement will continue to larger times. In order to do so one should compare the result with smaller dt to be sure that the results can be trusted.

Now this is not a minor issue, take for instance the problem I raised in the previous report to which the authors replied:

"We have tried the suggested procedure, i.e. plotting the excess energy versus C(t,0) and performing a 3-parameters fit (see attached figure CorrEn.pdf: (a) for 2-spin and (b) for 3-spin). However, the results remain more consistent with the strong ergodicity breaking scenario, without adding further insights. We thus prefer to not add them to the paper, in order to avoid overcharging it."

Let me note that the interest of such a plot is not to argue if favor or strong ergodicity breaking or not.
Indeed any claim on strong ergodicity breaking is based on extrapolations to infinite time and therefore prone to endless discussions, given that there is no analytical prediction on how the correlation with the initial condition should decay to zero. The aim of the parametric plot is essentially different, the question is: **assuming** that there is weak ergodicity breaking, does the asymptotic energy go to the threshold one? Looking from the figure attached to the reply one would say that this is not the case and this is a very interesting statement.
Therefore I urge the authors to put the figure in the manuscript (there are already 16 figures in the paper, I do not think that one more will change much in terms of readability).
However this is clearly an instance in which one would like to be sure that the data do not suffer from a systematic discretization error. Therefore I would like to see the same parametric curves with a different dt so that the data plotted can be taken confidently to describe the continuum dt=0 limit. The reader must know if the data they are seeing are reliable and this could be done in an appendix.

Requested changes

1) put the correlation-energy parametric figure in the text

2) display the effect of the dicretization dt on the data of the aforementioned figure in a separate figure, to be put in the appendix

  • validity: -
  • significance: -
  • originality: -
  • clarity: -
  • formatting: -
  • grammar: -

Author:  Giampaolo Folena  on 2023-06-10  [id 3722]

(in reply to Report 1 on 2023-05-24)

** Referee**

As I said in the previous report the paper deserves publication. However I ask the authors to make a small additional effort to address an important technical issue. The key ingredient of the paper is the numerical solution of the off-equilibrium dynamical equations by time discretization. Therefore the outcome depends on the time spacing dt. The correct procedure is then to extrapolate the solution to dt=0 considering different values of dt's. It seems that instead the authors report only data at fixed dt=.05 and do not mention the issue at all. Take for instance fig. 13 (a), it suggests that the numerical curve with dt=.05 does not have large corrections at least up to time t=100 due to the agreement with the results from the series expansion, BUT this is not enough to argue that the agreement will continue to larger times. In order to do so one should compare the result with smaller dt to be sure that the results can be trusted.

Response We agree with the referee that it is of fundamental importance to show that the chosen dt=0.05 gives a small error with respect to the exact dt -> 0 limit. Indeed this is the case, and we substantiated it in an additional appendix A, which provides explicit plots (Fig.13) of the error as a function of the time, showing that the error for dt=0.05 is small and decreases with time, and therefore it does not corrupt the analysis. We remark that that an analogous analysis of the error was provided in a previous article by one of the authors.

** Referee**

Now this is not a minor issue, take for instance the problem I raised in the previous report to which the authors replied:

"We have tried the suggested procedure, i.e. plotting the excess energy versus C(t,0) and performing a 3-parameters fit (see attached figure CorrEn.pdf: (a) for 2-spin and (b) for 3-spin). However, the results remain more consistent with the strong ergodicity breaking scenario, without adding further insights. We thus prefer to not add them to the paper, in order to avoid overcharging it."

Let me note that the interest of such a plot is not to argue if favor or strong ergodicity breaking or not. Indeed any claim on strong ergodicity breaking is based on extrapolations to infinite time and therefore prone to endless discussions, given that there is no analytical prediction on how the correlation with the initial condition should decay to zero. The aim of the parametric plot is essentially different, the question is: assuming that there is weak ergodicity breaking, does the asymptotic energy go to the threshold one? Looking from the figure attached to the reply one would say that this is not the case and this is a very interesting statement. Therefore I urge the authors to put the figure in the manuscript (there are already 16 figures in the paper, I do not think that one more will change much in terms of readability). However this is clearly an instance in which one would like to be sure that the data do not suffer from a systematic discretization error. Therefore I would like to see the same parametric curves with a different dt so that the data plotted can be taken confidently to describe the continuum dt=0 limit. The reader must know if the data they are seeing are reliable and this could be done in an appendix.

Response As suggested by the referee we have added a new plot of the energy vs the Correlation (Fig.8) and commented accordingly. We remark that the analysis of the error in dt provide evidence that the results shown in Fig.8 should have a 'very small' difference with the exact results in the dt->0 limit. For further confirmation we have added the extroplated (dt->0) line in Fig.8 for the (3+11)-spin model (dashed dotted red line) for comparison.

---

## Round 2 · Referee Report · Anonymous (Referee 1) · 2023-5-29

Report

I thank the authors for their response. I agree with reviewer 1 that a plot showing the impact of the time step dt on the extrapolation is important to support their finding. I believe that with this modification the paper is ready for publication.
  • validity: -
  • significance: -
  • originality: -
  • clarity: -
  • formatting: -
  • grammar: -

Author:  Giampaolo Folena  on 2023-06-10  [id 3723]

(in reply to Report 2 on 2023-05-29)

We have followed the suggestion of reviewer 1 and added an appendix which shows the impact of the time step dt .

---

## Round 2 · Author Response

We have carefully read and answered the two reports of our manuscript.
The two reports are positive and only a few small changes and precisations have been requested.
We have carefully considered all recommendations and we have modified the text accordingly.
We think that our manuscript is ready for publication.

---

## Round 2 · List of Changes

Report 1: - properly introduced the “overlap”, the “characteristic polynomial” and the “complexity”. - added footnote to explain that $\lambda$ is chosen to maximize the “discrepacy” - added comment on T_onset crossover versus transition.

Report 3: - added a footnote to comment about previous long time numerical solutions - rephrased the sentence to answer to the misquotation of [22] - clarified that in [54] the considered transition is of continuous type

---

## Round 3 · Referee Report · Anonymous (Referee 3) · 2023-7-6

Report

The Authors have satisfactorily addressed the issues I raised and I definitively recommend publication.

---

## Round 3 · Author Response

In response to the referees' suggestions, we have incorporated an evaluation of the magnitude of the errors in the numerical integration. We have made the necessary modifications to address the referees' requests and are confident that they have been successfully fulfilled.

---

## Round 3 · List of Changes

1) Added a new appendix that includes a description of the numerical integration algorithm together with a figure (Fig. 13) that presents an analysis of the errors' magnitude using a time step of dt=0.05. 2) Added Fig. 8 presenting the parametric plot of Energy vs Correlation and commented it in main text. 3) Changed convex -> concave in the appendix.

---

## Editorial Decision

published